# Evolving copy number gains promote tumor expansion and bolster mutational diversification

Zicheng Wang[1,2,3,7], Yunong Xia[1,2], Lauren Mills[4], Athanasios N. Nikolakopoulos[1,2], Nicole Maeser[1,2], Scott M. Dehm [1,2,5], Jason M. Sheltzer [6] & Ruping Sun [1,2,7] ✉

The timing and fitness effect of somatic copy number alterations (SCNA) in cancer evolution remains poorly understood. Here we present a framework to determine the timing of a clonal SCNA that encompasses multiple gains. This involves calculating the proportion of time from its last gain to the onset of population expansion (lead time) as well as the proportion of time prior to its first gain (initiation time). Our method capitalizes on the observation that a genomic segment, while in a specific copy number (CN) state, accumulates point mutations proportionally to its CN. Analyzing 184 whole genome sequenced samples from 75 patients across five tumor types, we commonly observe late gains following early initiating events, occurring just before the clonal expansion relevant to the sampling. These include gains acquired after genome doubling in more than 60% of cases. Notably, mathematical modeling suggests that late clonal gains may contain final-expansion drivers. Lastly, SCNAs bolster mutational diversification between subpopulations, exacerbating the circle of proliferation and increasing heterogeneity.

Underlying the maintained genomic diversity within a patient tumor is the uncontrolled proliferation, a core hallmark of cancer[1], coupled with somatic alterations occurring over time[2]. To prevent the disease, uncovering the somatic aberrations responsible for the malignant growth is the primary goal of precision oncology. At the genomic level, somatic alterations exist on a spectrum, ranging from small changes such as somatic single nucleotide variants (SSNV)[3] to large somatic copy number alterations (SCNA). Frequent chromosomal mis-segregation (chromosomal instability or CIN) leads to abnormal chromosome numbers (aneuploidy)[4] and unbalanced structural variations (SV) cause segmental SCNAs[5]. These two genomic errors are intertwined in many solid tumors, leading to extensive SCNAs, especially in advanced diseases[4] with poor clinical outcomes[6].

The inextricable relation of SCNAs to cancer initiation[7,8] and progression[9] has become widely recognized in cancer genomics. It remains little known, however, to what extent a specific SCNA accounts for the malignant growth and how it affects the intra-tumor-heterogeneity (ITH)[4]. Indeed, chaotic karyotype and widespread high copy number (CN) states in aneuploid tumors[10] pose a significant challenge in identifying oncogenic SCNAs, limiting the precision of using SCNA patterns for diagnostic and treatment purposes. For example, the treatment strategy for osteosarcoma, the most common bone tumor affecting teenagers with one of the most chaotic aneuploid genomes, has stagnated for decades[11]. From an evolutionary perspective, discovering the tempo of SCNA during somatic evolution is key to gaining knowledge of SCNA drivers[12]. Here, we hypothesize that the timing of SCNAs can be systematically measured from whole genome sequencing (WGS) data of patient tumors, and the temporal axis contains tangible information in isolating the effect of specific SCNAs on tumorigenesis.

[1]Department of Laboratory Medicine and Pathology, University of Minnesota, Minneapolis, MN, USA. [2]Masonic Cancer Center, University of Minnesota, Minneapolis, MN, USA. [3]School of Data Science, The Chinese University of Hong Kong (CUHK-Shenzhen), Shenzhen, China. [4]Department of Pediatrics, University of Minnesota, Minneapolis, MN, USA. [5]Department of Urology, University of Minnesota, Minneapolis, MN, USA. [6]School of Medicine, Yale University, New Haven, CT, USA. [7]These authors contributed equally: Zicheng Wang, Ruping Sun. ✉e-mail: ruping@umn.edu

We should pause to clarify how bulk sequencing data capture somatic evolution timeline. The tumor founder cell arose from the succession of clonal expansions in the pre-cancerous context where beneficial alterations endow progenitor cells with the ability to crowd out less advantageous populations[13] (Fig. 1A). The growth of the primary lesion gives rise to genomically diverging lineages[14], some of which, after acquiring a more malignant potential, can initiate the re-growth of a secondary tumor, such as metastasis[15]. Bulk sequencing data provide us with the opportunity to anchor the roots of expansion (the most recent common ancestor, or MRCA). Clonal variants in a single sample refer to the root of the observed sample. In multi-region sequencing, truncal variants from multiple samples could trace back asymptotically to the founder of the tumor[16]. For longitudinal sampling, e.g., of paired primary and metastatic tumors, truncal variants could point to the MRCA of the branched tumor progression[17]. Multi-samples reflect the population expansion at a broader scale, i.e., they coalesce to an earlier progenitor cell than a single sample does. Collectively, truncal variants revealed by a particular sampling strategy must map to the somatic evolution timeline prior to the corresponding sampling-relevant expansion (SRE).

The timing of a truncal SCNA on the evolution toward the MRCA could shed light on the impact of this variant on promoting the SRE. However, our knowledge about the SCNA timing remains fragmentary as the existing methods are restricted to simple (single or double)

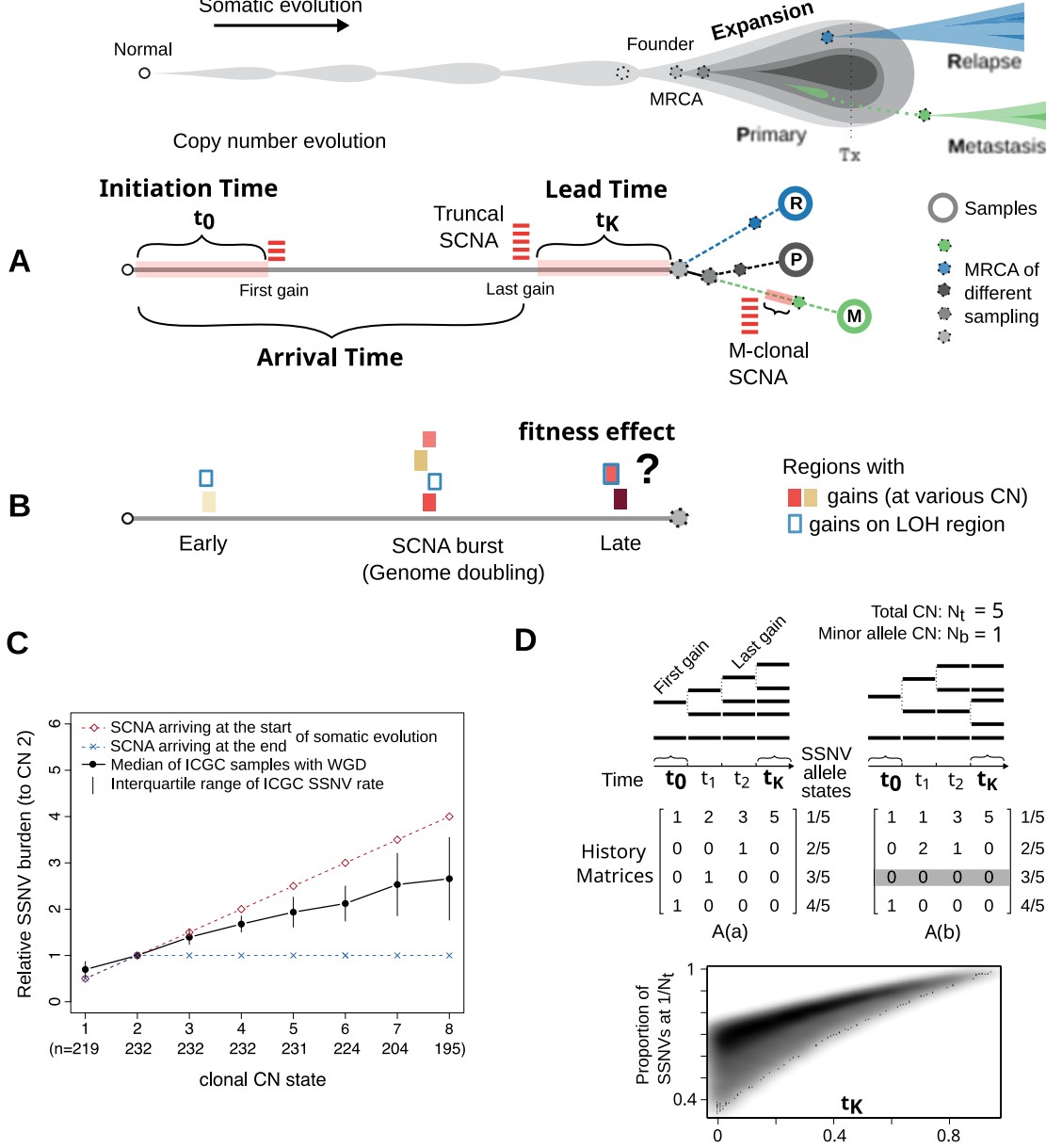

**Fig. 1 | Measuring the arrival and initiation time of SCNAs.** A drawing at the top marks the concept of cancer somatic evolution, which leads to the birth of the most recent common ancestor (MRCA) of the primary tumor, as well as the roots of secondary tumors. Tx: treatment. **A** For an SCNA present within the MRCA of a sampling-relevant tumor expansion, we aim to characterize the time when the last gain and first gain appeared (referred to as arrival and initiation time, respectively) for the corresponding genomic region. **B** We further aim to address the question if the late truncal gains are neutral or beneficial to tumor evolution. **C** The solid black line shows that the burden of SSNVs of a given genomic locus correlates with its CN state in tumors sequenced by ICGC (n: the number of independent samples with whole genome doubling). Two dashed lines assume the two extreme scenarios of SCNA arrival time. **D** The proportion of SSNVs at different allele states depends on the SCNA history matrix and the relative time span of each CN stage. Two possible history matrices are shown with the SCNA at 5:1. A 2D density plot in grayscale shows the burden of single-copy SSNVs against the lead time ($t_K$) simulated using the two matrices. Source data are provided as a Source Data file.

gains[16,18,19]. Single-sample based pan-cancer analysis from ICGC[19] have revealed the molecular time of these simple gains. Relative to aneuploid tumors, these low CN gains may not be sufficient to induce the final tumor expansion. For example, single chromosome gains have shown limited capability in driving proliferation in vitro[20]. As such, it would be crucial to know when a genomic segment further evolves beyond the simple gains, which can often culminate at a state greater than four copies[10]. This requires a timing method applicable to complex gains with high CN states.

In SCNA timing analysis, the following assumption is made: the site frequency spectrum (SFS) of SSNVs in a genomic region affected by SCNA depends on the trajectory (the order) and time span on each CN state that the segment had ever rested on[21]. For a single gain, the ratio between early (duplicated) and late (non-duplicated) SSNVs can be used to estimate the relative timing of the gain[12]. For a clonal SCNA involving multiple gains resulting in high CN (>4), one can divide its evolution timeline between germline to the founder of SRE into three fractions (Fig. 1A). The first time fraction ($t_0$) is the initiation time when the first gain occurs. The third fraction ($t_K$) is the lead time which measures the delay from the last gain to the onset of population expansion. We then define $1 - t_K$ as the arrival time. While the detailed trajectory is not identifiable from the SFS[18], we can still learn the upper bounds of these time fractions from the SSNV data. In particular, once a segment arrives at the final CN state, it can only accumulate single-copy SSNVs. The longer the segment persists in the observed CN state, the more overwhelming the single-copy SSNVs (Fig. 1D).

A significant question is how the timing of an SCNA reflects its impact on fitness (Fig. 1B). Whereas early gains could initiate and increase the risk of disease, propelling the initial proliferation of cancer cells, we suggest that late-appearing SCNAs close to MRCA could promote the population expansion more directly. If a clonal lineage persists over many generations and accumulates a significant number of mutations following the acquisition of an SCNA, it is improbable that the SCNA alone can drive the tumor's ultimate growth. Conversely, if an SCNA instigates the final expansion of a tumor, it is conceivable that the progenitor cell undergoes robust proliferation immediately upon acquiring the specific SCNA, resulting in few subsequent alterations attaining clonality. Punctuated acquisition of polyploidy (e.g., through genome doubling or GD) is prevalent in aneuploid tumors[22] but it remains unclear how close the occurrence of GD is to tumor transformation. Evidence exists that GD itself doesn't confer a strong fitness advantage[23]; instead, it can enhance the plasticity of the genome that permits further CN evolution, such that aneuploid cells adapt to overcome possible fitness penalties incurred by GD[24]. Therefore, SCNAs that arrive after GD could contain driver events. Moreover, depending on the precise location of biopsied tissue, single-sample analyses may differ in the corresponding time scale; subsequently, it is particularly essential to focus on the timeline toward the malignant growth - the somatic evolution in collecting truncal SSNVs of multiple samples of a tumor (e.g., multi-region samples or paired primary and metastatic samples).

In this study, aiming to broaden the "timeable" genomic regions for SCNAs, we develop `Butte` (BoUnds of Time Till Expansion), a computational framework to estimate the upper bounds of SCNA arrival and initiation time from WGS data. By applying `Butte` onto multi-sampled WGS data of five cancer types with widespread SCNAs, including osteosarcoma, we systematically chart the temporal patterns of CN evolution in vivo. To see if late-appearing SCNAs may confer fitness benefits, we construct mathematical models to examine the evolutionary mechanisms that give rise to these late truncal events. Furthermore, we also interrogate potential ways the late culminating SCNAs could add to the fitness and reveal its impact on mutational diversification during tumor expansion. The terminology employed in this manuscript is detailed in Supplementary Table S1.

## Results

### A computational framework to estimate the arrival time of SCNA gains

From the WGS data, one can characterize with high certainty the dominant SCNAs, inferring the integer allelic CN of a genomic region and the cellular prevalence of the corresponding SCNA, i.e., the percentage of cells sharing the dominant SCNA state[25]. We refer to a unique version of a genomic region (or segment) as an "allele". We term the total CN as $N_t$ and the CN for the minor allele as $N_b$ ("b" stands for b-allele determined by germline SNPs) for a dominant SCNA. The "allele" state ($a$) of an SSNV is the amount out of the total $N_t$ copies of the region that carry the corresponding variant. We found that in the aneuploid tumors sequenced by ICGC (International Cancer Genome Consortium)[26], the SSNV burden increases with the dominant SCNA states of the corresponding genomic region (Fig. 1C). The pattern can be attributed to an inherent correlation between SCNAs and SSNVs: a genomic segment resting on a CN state accumulates SSNVs at a rate proportional to the corresponding CN. Thereby, the burden and multiplicity of SSNVs are actively shaped by SCNAs.

The observed SCNA of a genomic segment (with a configuration $N_t$:$N_b$ different from 2:1) is the result of a series of CN events. For an SCNA involving at least $K$ gain events, the total time of somatic evolution can be divided into $K + 1$ stages. The segment begins with the 2:1 setting in the first stage and keeps "climbing up" by duplicating one of its existing copies in each subsequent stage, respectively, until it arrives at the observed SCNA state in the last stage (Fig. 1D). Accordingly, each stage is associated with certain time proportion ($t_k \geq 0$) and $\sum_0^K t_k = 1$, the total time for the somatic evolution. SSNVs occurring at stage $k$ on a segment copy that experiences duplication(s) in later stages will remain present on the duplicated copies with the allele state $a \geq 2/N_t$. By contrast, SSNVs acquired at stage $k$ on a copy without further duplications remain at the single allele state ($a = 1/N_t$). One can define a history matrix $A$ with entry $A_{jk}$ representing the number of segment copies in stage $k$ that produce the final allele state (i.e., frequency) $a_j = j/N_t$[18,21]. It can be seen that the abundance of SSNVs at allele state $a_j$ depends on $\sum A_{jk} t_k$. From the site frequency spectrum (SFS) of SSNVs in a region affected by SCNA, one can estimate the relative abundance of SSNVs at each allele state, and in turn, solve for each $t_k$. There has been much effort to infer $t_0$, i.e., the timing of the first copy number event[18,19]. These efforts focused on single gain (2:0 and 3:1) and at most double gains (3:0 and 4:1), where the history matrix $A$ is invertible. By contrast, for other SCNA states, multiple possible trajectories can exist and the underlying linear system is underdetermined, i.e., there are more time stages (unknown variables) than the possible allele states (equations). We note that, however, regardless of the underlying history, multi-allele SSNVs ($\geq 2/N_t$) can only occur before the last stage ($K$) of CN evolution; once the genomic region arrives at the observed clonal SCNA state, all the copies ($N_t$) would accumulate SSNVs at single allele state ($1/N_t$). Therefore, the longer the last stage of CN evolution (from the emergence of the SCNA to the onset of population expansion), the more overwhelmingly the single allele SSNVs dominate the SFS (Fig. 1D). The monotonicity property enables the examination of the proportion of time of stage $K$.

To investigate how various SCNAs unfold during somatic evolution, we developed `Butte` (BoUnds of Time Till Expansion), which adopts linear programming to infer the upper bounds of lead ($t_K$) and initiation time ($t_0$) of SCNAs (Fig. 2). `Butte` extends the full maximum-likelihood estimation procedure implemented in `cancerTiming`[18]. Notably, `Butte` does not restrict the analysis to single and double gains, but in addition allows the calculation of the upper bounds of $t_K$ and $t_0$ for SCNAs up to seven total copies, broadening the "timeable" SCNA regions. The estimated timing systematically correlate with the actual ones of SCNA initiation and culmination (Supplementary Fig. S1). To see the effect of SCNA history, read depth ($D$), number of SSNVs ($M$) and tumor purity ($P$) on the timing inference, we simulated

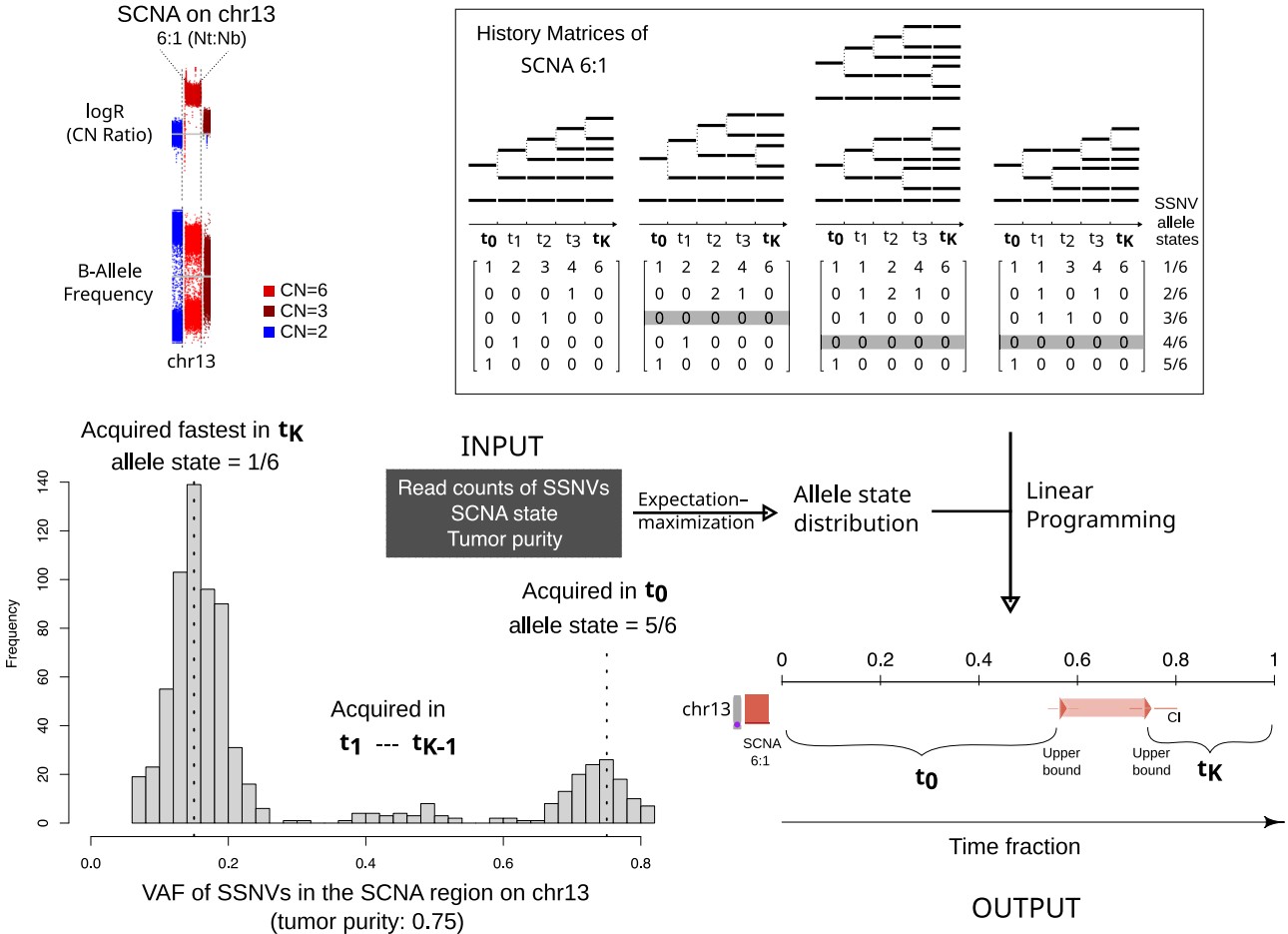

**Fig. 2 | An example workflow of** `Butte`**.** Tumor ESCA_R_8 has a clonal SCNA (at 6:1) on chr13, as indicated on the top left. LogR is the Log2 copy number ratio between the tumor and matched normal sample and the B-Allele Frequency quantifies the allelic imbalances. `Butte` takes the read counts (total depth and the depth of the mutant allele) for SSNVs on chr13, the SCNA state and tumor purity as the input. `Butte` then works out the allele state distribution by using Expectation Maximization. By adopting linear programming with all possible history matrices, `Butte` returns the upper bounds of initiation and lead time of the SCNA, respectively (CI: confidence interval). The variant allele frequency distribution of SSNVs is shown on the bottom left to illuminate the relationship between allele states and SCNA timing. Note that the VAF is affected by tumor purity.

different histories for SCNAs at 6:1 and 6:2, respectively and obtained virtual SSNV data (see Methods, Supplementary Figs. S2, S3, S4, S5). $M$ have a higher impact on timing accuracy than $D$ and $P$. While the overestimation of $t_K$ and underestimation of $t_0$ is prominent when $M$ is less than 20, the timing results are robust for $M$ above 50, especially for SCNA 6:1. As SCNA 6:2 involves gains for both alleles, the resulting SSNVs have a smaller set of possible allele states than SCNA 6:1. For 6:2, `Butte` tends to overestimate $t_K$ when it is small ($\leq 0.3$). Conversely, $t_0$ can be underestimated if the intermediate stages are brief (e.g., $t_0 \geq 0.5$

and $t_K = 0.3$), owing to the penalty for infeasible models implemented in `Butte`.

To test the performance of `Butte` on real tumors, we first evaluated the timing predictions by analyzing multi-region WGS data of colorectal adenocarcinomas (COAD)[16,27]. Similar to the simulation results, a higher number of SSNVs leads to increased precision in timing estimation (Supplementary Fig. S6). With 50 SSNVs, the median confidence interval falls below 0.2. `Butte` successfully identified early CN gain of chromosome (chr) 5q (Supplementary Fig. S7), corresponding to the SCNA state of 2:0 (copy neutral loss of heterozygosity), a known early step in COAD initiation involving gene *APC*[28]. Additionally, `Butte`'s event timing on the ICGC BRCA dataset[29] aligns with a method employing graph theory to predict genome rearrangement history using both SSNVs and SVs (Supplementary Fig. S8). The timing of low copy number gains estimated by `Butte` corresponds with predictions made by the method emphasizing joint likelihood of copy number timing[16] (Supplementary Fig. S9). As a reference for late-appearing events, private (sample-specific) SCNAs should contain events that arise in the descendent lineage of the MRCA of multi-samples. Using all the multi-sampled WGS data listed in Table 1, `Butte` predicted their arrival time to be later than the public SCNAs on the timeline leading to the MRCA. This underscores its ability to uncover SCNAs that occur at a later stage. (Supplementary Fig. S10).

## Table 1 | WGS data included in this study

| Tumor | Refs. | Accession Code | Sampling[a] | #Samples[b] | #Patients |
|---|---|---|---|---|---|
| OS | 30 | EGAD00001004482 | MTS | 17 | 9 |
| | 31 | EGAS00001000263 | Single,MTS | 24 | 22 |
| COAD | 16 | EGAD00001004966 | MRS | 43 | 7 |
| | 27 | phs001722.v1.p1 | MRS,MTS | 7 | 2 |
| BRCA | 32 | EGAD00001002696 | MTS | 26 | 12 |
| | 33 | JGAD000095 | Single | 10 | 10 |
| PRAD | 35 | EGAD00001000891 | MRS,MTS | 47 | 9 |
| ESCA | 34 | EGAD00001001394 | MRS | 11 | 4 |

[a]*MRS* multi-region sampling, *MTS* multi-tumor sampling.
[b]Samples passed our quality assessment (Supplementary Figs. S23, 24 and 25).

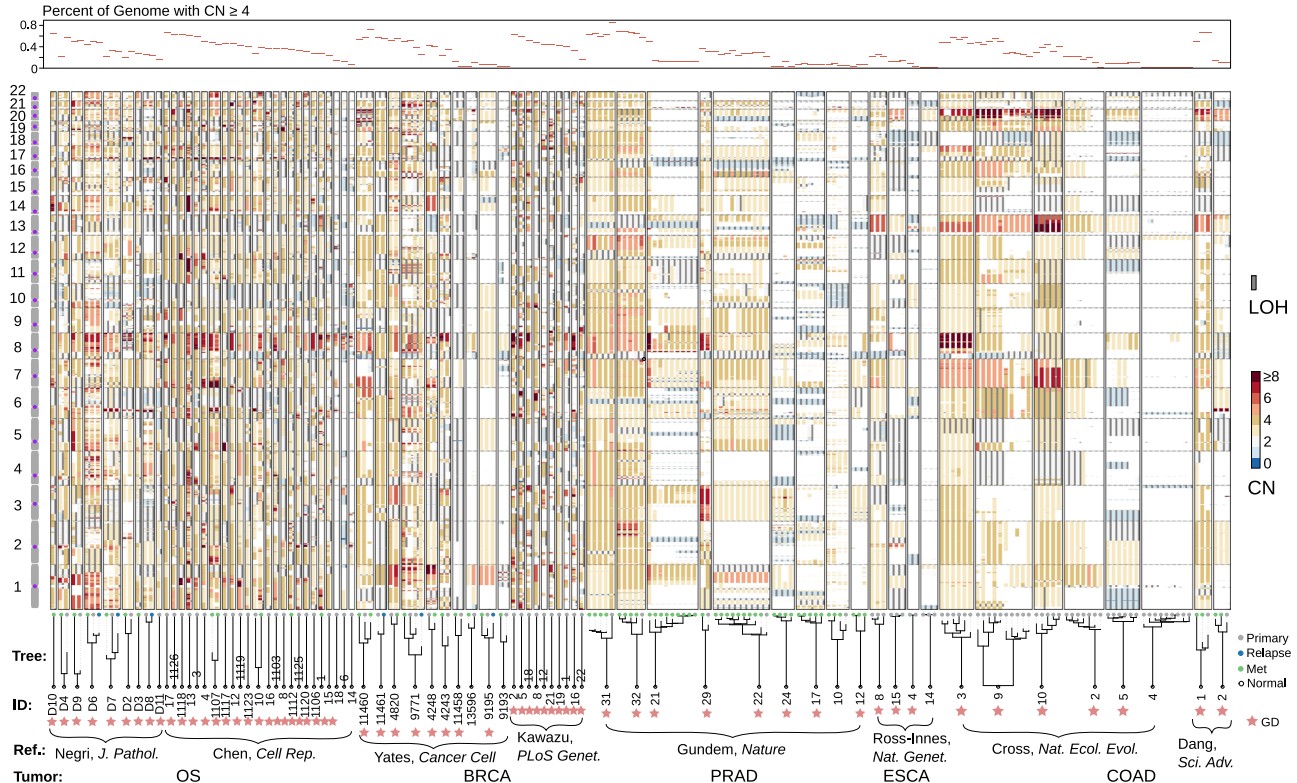

**Fig. 3 | CN profiles across five tumor types from the re-analysis of published WGS data.** Vertical bars represent the segmental CN states along the autosomal chromosomes characterized by the WGS data of tumor samples. For each sample, a color-coded thick bar shows the total CN state of each genomic locus and a thin gray bar to the right indicates that region has loss of heterozygosity (LOH). Samples belonging to the same patient are boxed. The top panel highlights the fraction of high CN states in each sample's genome. The lower panel exhibits the sample phylogenetic trees constructed from SSNVs. Sample IDs, the reference where the WGS data was published, and tumor types are tabulated at the bottom. Presence of GD are indicated. In this manuscript, a tumor sample is named after the concatenation of the tumor type, the first character of the author's surname and the patient ID. Source data are provided as a Source Data file.

To guarantee accuracy in identifying late and early gains using a predefined time threshold, we evaluated the precision, false positive, and true positive rates for predicting $t_O$ and $t_K$, respectively (see Methods for details). Overall, these evaluations showed an increase in performance metrics with the threshold values (Supplementary Figs. S11–S12). To strike a balance between precision and false positive rate, we defined late gains as those occurring in the last 20% of the truncal time measured by clonal SSNVs, and early gains as those occurring in the first 20%.

### Evolving SCNA gains define the tumor transformation leading to the most recent clonal expansion

To evaluate the tempo of SCNAs in solid tumors, we applied `Butte` on five tumor types by analyzing eight published WGS datasets: osteosarcoma (OS)[30,31], breast invasive carcinoma (BRCA)[32,33], colorectal adenocarcinomas (COAD)[16,27], esophageal carcinoma (ESCA)[34], and prostate adenocarcinoma (PRAD)[35], six of which comprise multi-sampling of patient tumors (Fig. 3, Table 1). 70% of the analyzed genomes (corresponding to 87% of the patients) were near triploid, with the median fraction (IQR) of the high amplitude CN regions (≥4) being 0.37 (0.23 to 0.49). Loss of heterozygosity (LOH) is prevalent but mostly is at copy neutral or amplified states in the triploid tumors. High amplitude gains can be recurrent across cancer types (e.g., chr 8q) or within a specific tumor type (e.g., chr 1q for BRCA, chr 17p for OS, and chr 7 for COAD). These recurrent gains presumably contain driver events[36], yet their tempo in somatic evolution remains uncharted. Notably, karyotypes largely remain stable across different samples of the same tumor, despite the presence of continued subclonal CN diversification in a relatively minor fraction of the genome.

We note that 74 out 75 patient tumors acquired late-appearing gains close to MRCA regardless of the overall ploidy or tumor type (Fig. 4A, B), with the only exception of COAD_C_4, which shows high microsatellite instability. Punctuated copy number bursts were observed in the triploid samples, reflecting the ability of the genome to leapfrog over intermediate states to reach moderately high CN states through whole or partial genome doubling (GD)[22,37]. Most of them could derive from whole genome doubling, but we could not exclude the possibility that individual tumors had duplications of multiple chromosomes instead of the whole genome. Here we refer to these synchronized gains as GD. Whereas GD occurs late (close to the MRCA) in some adult cancers (18 out of 34 patients), it appears to be an earlier event in many other tumors. This is particularly evident in OS where 28 of 30 patients had GD at the mid-stage of somatic evolution toward SRE (Fig. 4B). The contrasting tempo of GD suggests that it probably has a context-dependent function. In tumors with early GD, `Butte` can characterize the post-GD CN evolution, whereby progenitor cells continue to sample the aneuploid fitness landscape[24]. Notably, the rate of gains is higher post-GD than the rate pre-GD (paired Wilcoxon test, two-sided, V value = 630, sample size = 35, $p$ = 5.8e-11, effect size Cohen's d = 1.13, 95% Confidence Intervals of the effect size ranges from 0.62 to 1.65, Supplementary Fig. S13). Such an SCNA evolution involves stochastic chromosomal or structural abnormalities; however, certain genomic regions preferentially exhibit late gains across different patients in a particular tumor type, which, includes those recurrent high amplitude gains, such as chr 8q in OS (Fig. 4C) and chr 7 in COAD (Supplementary Fig. S7). On the other hand, recurrent SCNAs appear to initiate early, e.g., chr 1q in BRCA, chr 8q and chr 17p

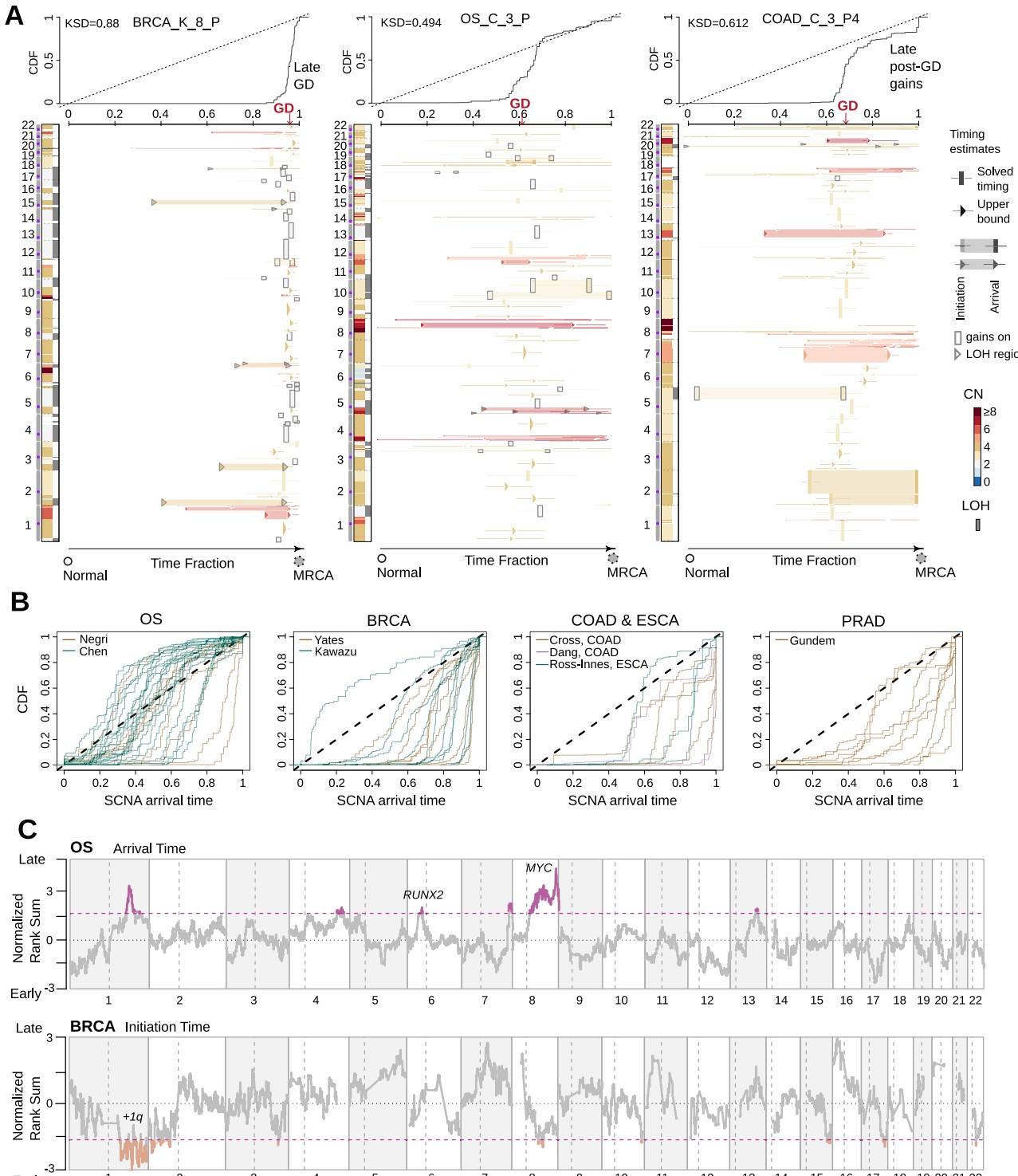

**Fig. 4 | The timing patterns of SCNAs across five tumor types. A** SCNA timing of three exemplified tumors. CN states along the genome are shown on the left of each panel. The right panel visualizes the time fraction of somatic evolution from germline to the MRCA of the patient tumor. For each SCNA segment, the inferred time points for its initiation and arrival are shown as either rectangles (exactly solved timing) or arrows (upper bounds of timing when the solutions are not unique) with the same color-coding as its CN. Confidence interval of the inferred timing is drawn by lines. The top panel shows the cumulative distribution (CDF) of SCNA arrival time. **B** The CDF curve of SCNA arrival time is shown for each patient categorized by the tumor type. **C** The figure displays normalized rank sums of timing across patients for each genomic bin, representing initiation time for BRCA and arrival time for OS (see Methods). Color-highlighted bins indicate recurrent early-initiating gains for BRCA and recurrent gains established late for OS (with 90% confidence level). Source data are provided as a Source Data file.

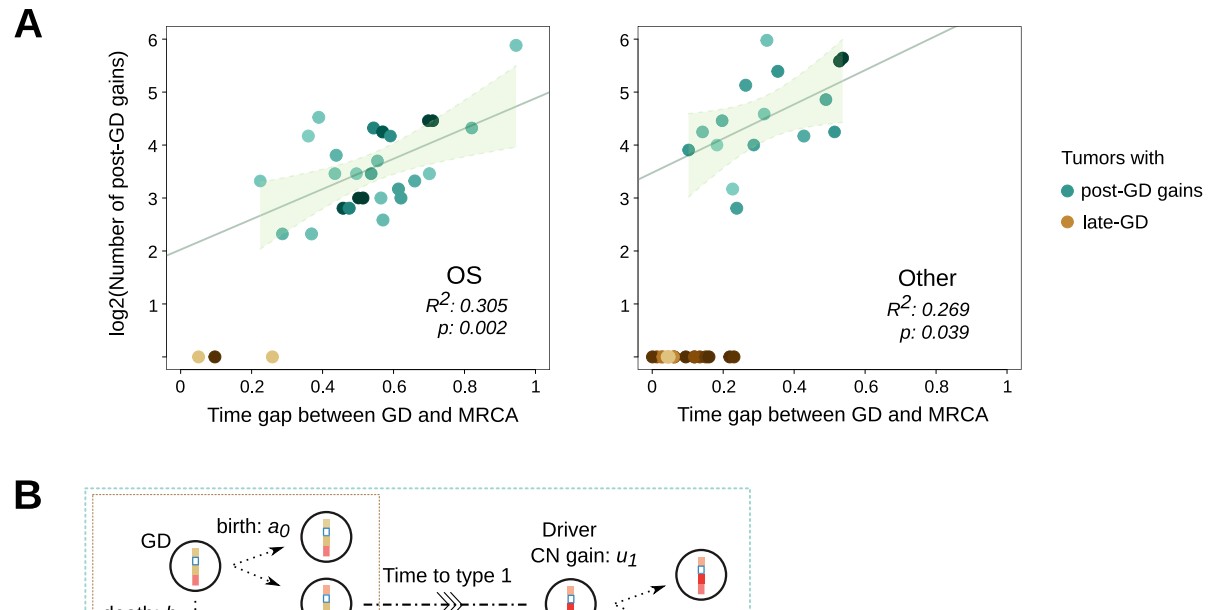

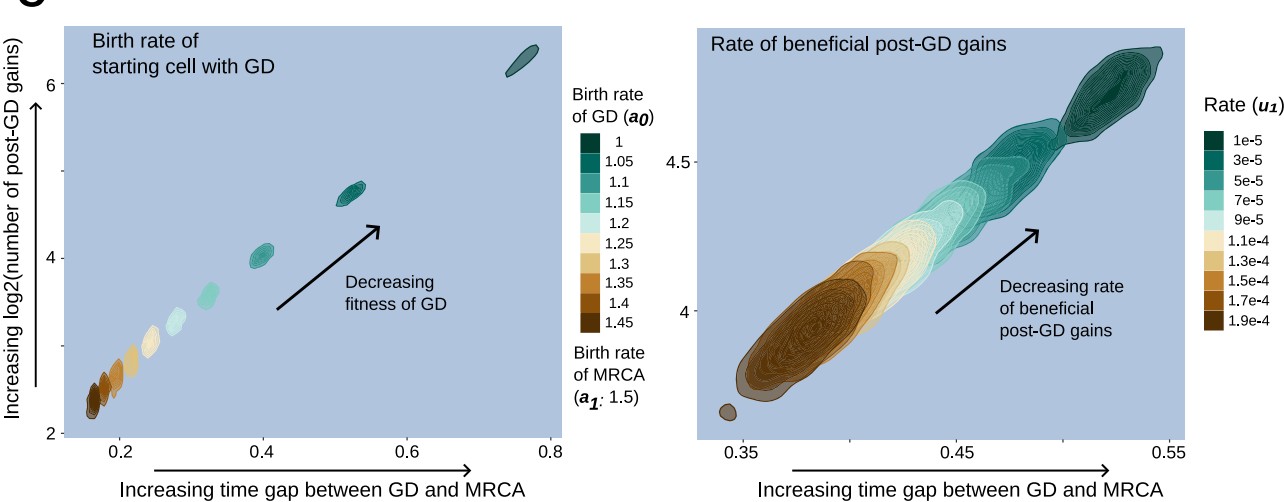

**Fig. 5 | Mathematical modeling suggests that final-expansion driver gains should occur late. A** Scatter plots colored by density illustrate the number of post GD gains (in log2 scale) against the time fraction of post-GD evolution to MRCA for OS and other adult cancer types, respectively. The green center line represents the fitted regression line (for tumors with post-GD gains) and the the green error bands represent the 95% confidence interval limits (with $R^2$ and $p$ values as noted). **B** The schema shows the setup of the two contrasting mathematical models: (1) GD is followed by neutral evolution where additional gains do not confer a fitness advantage and (2) post-GD gains increase the growth rate. **C** 2D density plots of the two metrics as in panel 5A characterized by the selection model. We studied the effects of the growth rate of GD (with a fixed growth rate of the

MRCA, the left panel) and the rate of beneficial post-GD driver gains (the right panel), respectively. We simulated 10,000 cases for each parameter setting. The density is produced from smoothed data points with each point referring to an average of 50 cases. To convert the post-GD evolution time in the simulation into fractions as in (**A**), we assume that GD occurs at 120 time units (roughly corresponds to a GD rate of $7 \times 10^{-5}$ [82] during pre-GD evolution with a birth and death rate at 1.1 and 1, respectively). Note that the modeling is not intended to infer the parameters from patient data. Left panel: $b_0 = 1$, $a_1 = 1.5$, $b_1 = 1$, $u_0 = 0.2$ [45], and $u_1 = 0.00001$; right panel: $a_0 = 1.05$, $b_0 = 1$, $a_1 = 1.5$, $b_1 = 1$, and $u_0 = 0.2$. Source data are provided as a Source Data file.

(*TP53*) in OS and chr 5q (*APC*) in COAD (Fig. 4C and Supplementary Fig. S7). Recent experiments have highlighted a significant fitness advantage linked to chr1q gains[38]. It is noteworthy that while these recurrent SCNAs tended to emerge either earlier or establish later in comparison to less common SCNAs, the correlation between copy number and timing was not consistent across all genomic regions

(Supplementary Fig. S14). These additional gains pre- and/or post-GD could result from either high evolvability of the corresponding region, or persistent selection upon driver genes within.

The earlier the timing of GD, the more post-GD CN gains (Fig. 5A, where OS and other adult tumor types exhibit $R^2$ values of 0.31 and 0.27, F statistics of 11 and 5.2, degrees of freedom of 25 and

14, and *p* values of 0.002 and 0.039, respectively). The late evolving gains are shorter in segment length than those associated with GD (Supplementary Fig. S15). This suggests that the post-GD CN evolution is driven by SVs, which occur at a higher rate than chromosomal mis-segregation. Indeed, the breakpoints of structural variants almost locate at the boundary of SCNA segments (Supplementary Fig. S16). As SVs continued to occur, regions containing driver genes could become focally amplified due to selective advantages. These genes would thus appear more often in the late gains, e.g., *MYC*[39] and *RUNX2*[40] in OS (Fig. 4C). Given the same total copy number, amplified LOH ($N_b = 0$ and $N_t \geq 3$) tend to culminate later than other types of amplifications, such as allele specific gains ($N_b = 1$ and $N_t \geq 3$). For example, at $N_t = 5$, amplified LOH (53 events) exhibits a longer arrival time than allele specific gains (106 events). The one-sided Wilcoxon test yielded a W statistic of 3656, a *p*-value of 0.001, and an effect size Cohen's d of 0.57. The 95% Confidence Intervals for the effect size range from 0.23 to 0.9. This contrast cannot be explained by the overestimation of `Butte` (Supplementary Figs. S17 and S1). Whereas truncal LOH were supposedly acquired before GD[15] causing the complete loss of tumor-suppressor activity, the late appearing gains of the only remaining allele may indicate that these regions potentially acquire dosage-dependent gain-of-functions[41].

## Mathematical modeling suggests the role of late truncal gains in promoting final expansion

While early genomic alterations garner significant attention for their functional implications, our understanding of the significance of late truncal alterations that emerge in proximity to the MRCA remains limited. Employing mathematical models, we constructed a rationale mentioned in the Introduction: a truncal alteration leading to the final expansion might be situated near the MRCA if the progenitor cell undergoes rapid proliferation upon acquiring this specific alteration. Using GD as an example, as it can occur early or late, we reason that early GD and prior alterations alone are insufficient to drive the final tumor expansion. However, if GD occurs closer to the MRCA, it, along with other late aberrations, can propel the tumor's ultimate growth.

We utilized a multi-type branching process model[42] to simulate tumor somatic evolution, focusing on two simplified models for key insights applicable to complex contexts. In the base model (neutral model), initiated with a single tumor-initiating cell acquiring the first driver mutation (GD, Fig. 5B), cells reproduce at a rate of $a_0$ and die at a rate of $b_0$, resulting in net growth rate $\lambda_0 = a_0 - b_0 > 0$. Post-GD, daughter cells acquire passenger mutations at a rate of $u_0$ without altering the net growth rate. In this context, tumor expansion is solely driven by GD, and all post-GD gains are passengers. Cells lacking beneficial post-GD gains are type 0 cells. In the selection model, cells with GD can acquire beneficial post-GD gains at a rate of $u_1 < u_0$, enhancing fitness ($\lambda_1 = a_1 - b_1 > \lambda_0$, detailed in Methods). Our goal is to characterize post-GD gains reaching fixation or dominance in tumors under two scenarios: without and with beneficial post-GD gains. Notably, in both models, post-GD gains are proportional to the mutation rate and time between GD and MRCA, validated partially in (Fig. 5A).

We first analyze the base model. Conditioned on the non-extinction of the population, we can obtain that the number of post-GD gains reaching fixation follows a geometric distribution with parameter $\frac{\lambda_0}{\lambda_0 + u_0}$ and mean $\frac{u_0}{\lambda_0}$ (see Methods). The mode of this distribution is at zero, similar to the cases where GD appears late and post-GD CN gains are rare. To tolerate the inclusion of subclonal but dominant SCNAs as the clonal variants, we further evaluated the dominant post-GD gains shared by the majority ($\geq 90\%$) of cancer cells. Building on the results of[43], we derived the expected number of

dominant post-GD gains in a tumor with size $N$ as

$$\tilde{S} = \frac{N}{\lceil 0.9N \rceil} \cdot \frac{u_0}{\lambda_0} \approx 1.11 \frac{u_0}{\lambda_0}, \quad (1)$$

which is only slightly larger than the clonal ones. Assuming that $u_0$ and $\lambda_0$ are comparable (based on experimentally measured $u_0$ for SCNA around 0.2 and the cancer cell death rate not significantly approaching the birth rate[44,45]), $\tilde{S}$ would be no more than just a few. Moreover, numerical simulations show that the number of dominant post-GD gains continues to follow a geometrical-like distribution with the mode at zero (Supplementary Fig. S18). Thus, if post-GD gains do not provide growth benefits, GD would be one of the last events before the MRCA as few of post-GD gains can become dominant in the observed tumor.

However, if cells with GD can acquire an additional beneficial post-GD gain (meaning GD alone is not enough to drive the final expansion), the situation drastically changes. To emphasize how this happens, let us denote cells with the beneficial post-GD gain by type 1 cells and consider the first type 1 cell that grows into an infinite number of descendants. We assume that the descendants of the first type 1 cell dominate the cell population when the sampling is performed (see Methods for details). The expected number of passenger post-GD gains ($\bar{S}$) carried by a type 1 cell at the moment of its introduction is proportional to the time of occurrence of the type 1 cell (the birth time of the tumor-initiating cell is set to be 0). In Methods we show that the distribution of the birth time of the first non-extinct type 1 cell, $\mathbb{P}(\sigma_1 > t | \Omega_\infty)$, where $\sigma_1$ represents the birth time and $\Omega_\infty$ represents the event that the population does not go extinct, can be characterized as a function of the rate of beneficial gains $u_1$ and growth parameters of type 1 and type 0 cells, respectively (Lemma 1). $\bar{S}$ is thus

$$\bar{S} = \int_0^\infty \mathbb{P}(\sigma_1 > t | \Omega_\infty) u_0 dt, \quad (2)$$

where we utilized the formula (see Section V.6 of[46]) to calculate the expected birth time of the first non-extinct type 1 cell. This calculation involved utilizing the tail probability ($\mathbb{P}(\sigma_1 > t | \Omega_\infty)$) and multiplying the expected birth time by the passenger mutation rate $u_0$.

We explored various choices of growth parameters that capture the fitness difference between type 0 (without beneficial post-GD gain) and type 1 cells (with beneficial post-GD gain). As compared to the base model, the selection model results in a much higher abundance of post-GD gains across a large parameter space (Supplementary Fig. S19). Notably, lowering the fitness level of type 0 cells delays the birth of the type 1 cell (Fig. 5C), conditioned on a fixed net growth rate of the type 1 cell. The prolonged period of post-GD evolution (accompanied by a higher abundance of post-GD gains) could also be attributable to a lower rate of beneficial post-GD gains (Fig. 5C).

If the identified SCNAs in patient data represent the dominant clone of the entire tumor (a scenario more likely in multi-sampling than single-sampling), our model implies that late clonal gains may harbor drivers for final expansion. The presence of early GD in many patient tumors suggests that post-GD gains might offer added advantages for promoting the final expansion. It's worth noting that our mathematical model doesn't rule out the possibility of late somatic alterations of other types, beyond copy number gains, driving the final expansion. Thus, investigating late alterations in various forms of somatic changes is an area of interest.

In the context of multiple drivers, previous research[28] indicates that the driver conferring the highest fitness advantage is likely to emerge early in random occurrences. Our simulations, shown in Supplementary Fig. S20, support this idea. When two drivers occur at equal rates, the one with the greater fitness increase is more likely to emerge first in the initial cell acquiring both drivers. Additionally, the probability of this early appearance increases with the magnitude of

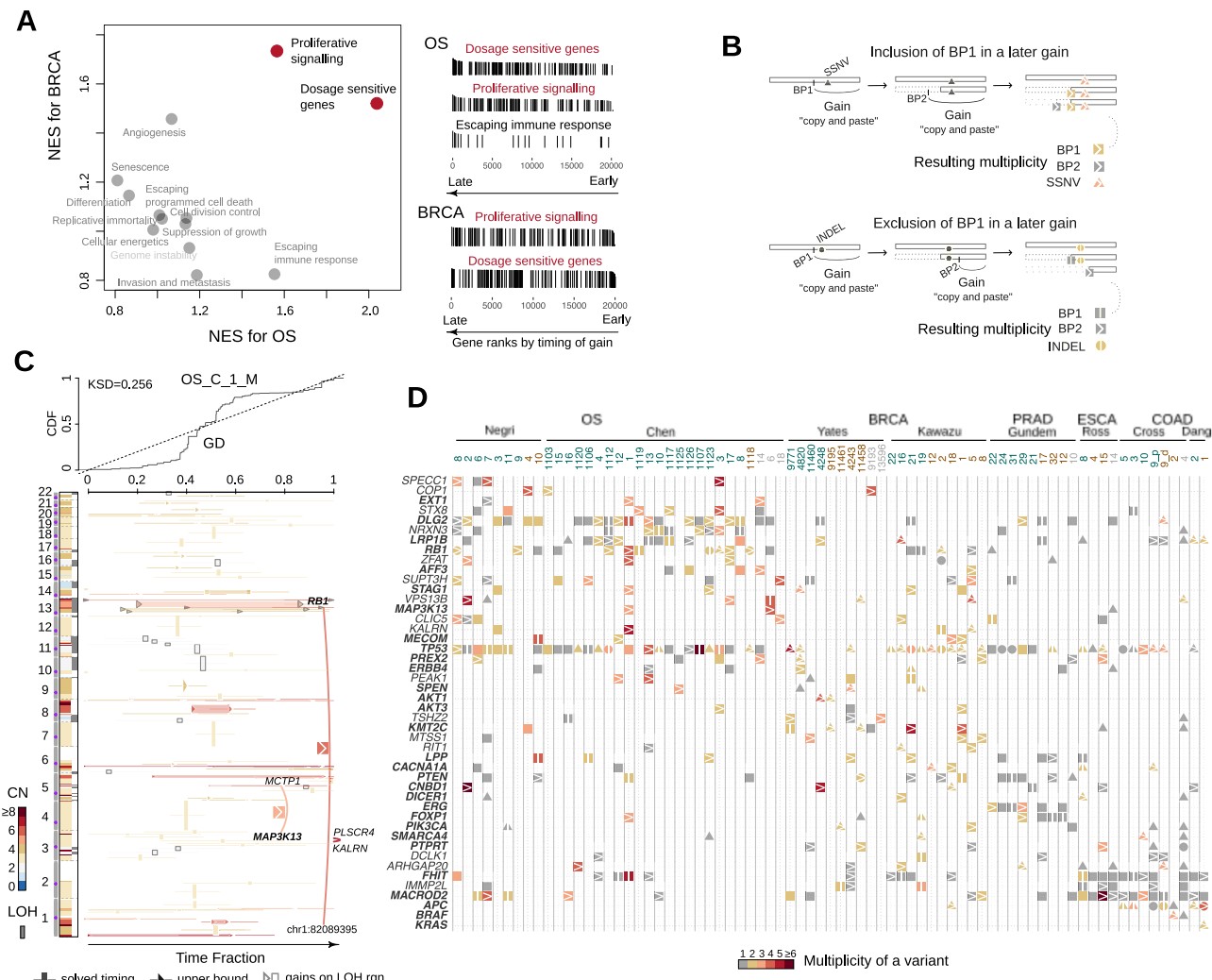

**Fig. 6 | Ways the late CN gains contribute to the fitness of cancer progenitor cell. A** The gene set enrichment analysis (GSEA) was performed on the gene list ranked by the averaged CN arrival time for BRCA and OS tumors, respectively. The scatter plot on the left shows the normalized enrichment scores (NES) for each set of cancer census genes belonging to the predefined cancer hallmarks by COSMIC database. Red-colored highlighted pathways have a false discovery rate less than 0.15. The vertical bars on the right panels visualize the timing-ranks of genes that belong to the highlighted gene sets. The height of each bar corresponds to the arrival time. **B** A cartoon illustrates the multiplicity increase of an early sequence variant due to the inclusion of that variant by a late CN gain, with annotations indicating the type of variants (symbol shapes), level of multiplicity (color hues) and the variants' association with a late gain (right arrow) or an early gain (vertical bar). BP: breakpoint. **C** The SCNA timing plot of an example OS tumor similarly arranged as in Fig. 4, with additional links and symbols highlighting the SV breakpoints in known cancer genes that are amplified by late gains. **D** The matrix plot demonstrates genes with recurrent somatic variants and their multiplicity across the five tumor types. Note that a high multiplicity indicates that an early somatic variants gained more copies of the mutant. Names for known cancer genes are in bold. Genes with variants showing higher multiplicity levels than gene *TTN* are also included. Symbol annotations are the same as in (**B**). Source data are provided as a Source Data file.

the fitness advantage provided by the more beneficial driver. These findings suggest that early gains may involve drivers with significant fitness effects, while late truncal gains near the MRCA are particularly relevant for the fitness increase driving the final expansion.

**Ways evolving CN gains contribute to fitness increase and mutational diversification**

As SCNAs have a global impact on gene expression in cancer[47], the evolving CN gains potentially affect dosage-sensitive genes whose gains have a functional impact. In the OS and BRCA tumors, as the CN evolves, we can see an enrichment of putative dosage-sensitive genes that are in pathogenic CNV peak regions derived from dbVar[48,49] (Fig. 6A). Moreover, we observed a similar enrichment for genes involved in sustaining proliferative signaling: one of the most fundamental capabilities of cancer cells[1]. No such enrichment is observed when utilizing copy number (CN) data alone

(Supplementary Fig. S21), underscoring the valuable additional insights provided by incorporating timing information. *MYC*, *EGFR* and *KIT* are among such genes with late gains in both OS and BRCA, emphasizing their ability in stimulating cell multiplication in multiple tumor types. In addition, post-GD late gains tend to affect genes whose inactivation (upon CRISPR knockout) alters cell proliferation dynamics[50] according to the DepMap database (Supplementary Fig. S22).

The evolving gains could amplify the impact of early functional variants by increasing their multiplicity (Fig. 6B). Such a mechanism potentially affects SV breakpoints in known oncogenes (e.g., *MAP3K13*, *MECOM* and *PREX2*), breakpoints in genes known to be involved in oncogenic fusions (e.g., *AFF3*, *LPP* and *ERG*), and simple mutations in oncogenes (e.g., SSNVs in *SMARCA4* and *CACNA1A*), see Fig. 6C, D. *MAP3K13* had been shown to promote tumor growth in high *MYC*-expressing cells[51,52], a similar context as in the OS[39].

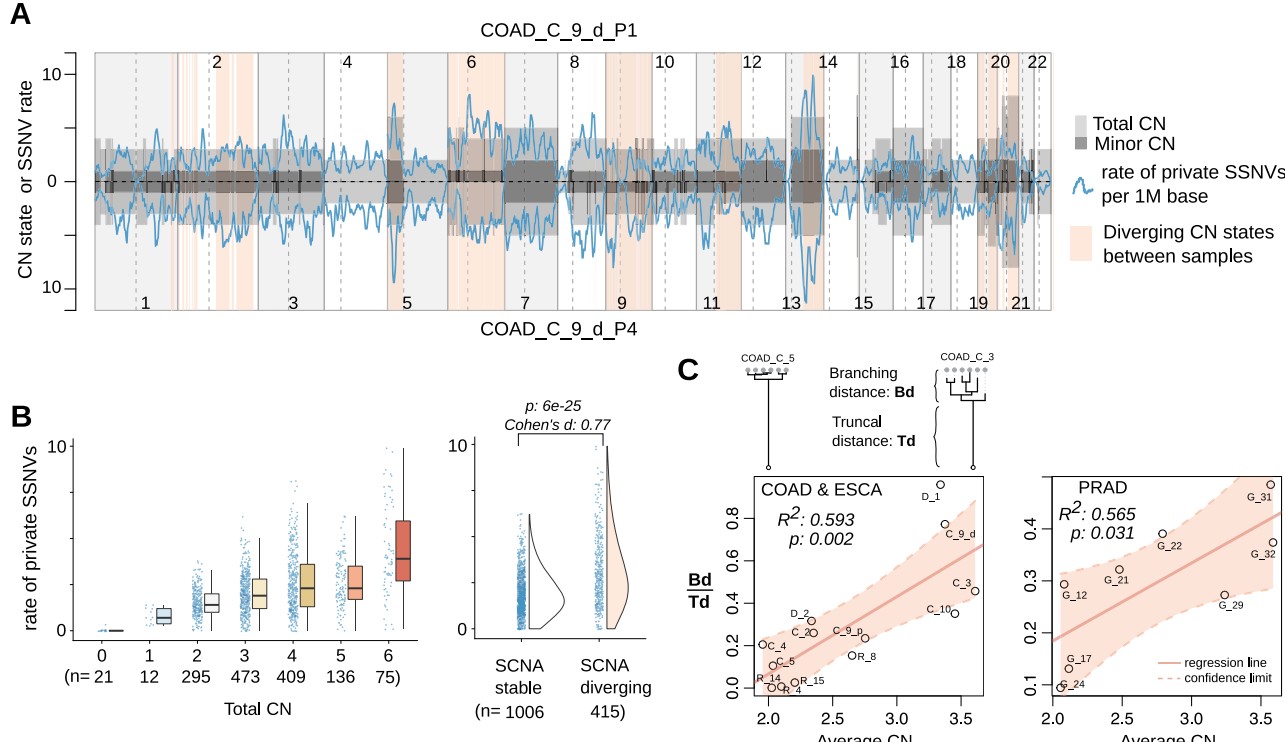

**Fig. 7 | The effect of SCNA on SSNV diversification during tumor expansion.**
**A** The rate of sample-private SSNVs when comparing two samples of a COAD patient tumor. The segmental CN states (total and minor CN) along the autosomal chromosomes for the two tumor samples are shown as gray rectangles above and below the x axis. The rate of sample private SSNVs (per million base pairs, blue line) fluctuates with the CN states, supporting the assumption that point mutations accumulate at a rate proportional to their CN. Genomic regions with different CN states between the two samples are in light red background. **B** The box plots on the left panel illustrate the rate of private SSNVs in sample P1 detected in regions at a given total CN state. The box represents the interquartile range, covering the central 50% of the data. The line inside the box indicates the median. Whiskers extend to the minimum and maximum values within a specified range, excluding outliers. The half-violin plots on the right panel demonstrate such a rate for regions showing stable or diverging CN states between the two samples with $p$ value of Wilcoxon rank sum test (two-sided, W value = 136214 and 95% confidence interval ranges from 0.7 to 1) and effect size indicated. **C** The branching distance relative to truncal distance in a tumor's phylogenetic tree was calculated for each of the COAD, ESCA and PRAD tumors to evaluate the correlation with the averaged CN of the corresponding tumor samples. Annotations show the percentage of variance explained by a linear regression model (with $p$ values of the model fitting shown). Source data are provided as a Source Data file.

We note that highly mutated tumor suppressor genes (TSG), such as *TP53*, *RB1* and *APC*, also have their early mutants duplicated or amplified (Fig. 6D). Whereas these are presumably inactivation variants, the retaining of multiple copies of the variants could suggest different roles that remain unclear, such as a potential gain-of-function of APC mutants in COAD[53]. The fact that SRE requires the amplifications of some early variants, rather than starting immediately upon acquiring a single copy of these variants, suggests that late-appearing gains could cooperate with the early variants to promote tumor expansion. On the other hand, late SV breakpoints (at single copy state) coupling evolving gains are prominent in genes located in common fragile sites, e.g., *FHIT* and *MACROD2*. Late alterations of these genome "caretakers" could facilitate further genome evolution and expedite clonal expansion[54,55].

Lastly, the quantitative relation between SCNA evolution and SSNV accumulation, the rationale behind our timing method, implies that SCNA gains bolster mutational diversification between subpopulations during tumor growth. In principle, the higher the truncal CN state of a genomic segment, the higher the mutational divergence between subclones for the corresponding locus. As tumor expands, genomic regions at distinct SCNA states would accumulate SSNVs at different rates, leading to the heterogeneity of the SSNV burden along the genome. For example, when comparing two samples of a tumor, the sample-specific SSNVs are more abundant for regions with higher CN states (Fig. 7A, B). Notably, the overall CN state affects the structure of phylogenetic trees, i.e., it explains more than 50% of

the variance of the relative branching distance measured by SSNVs in COAD and PRAD patients, where extensive multi-region sampling is available (Fig. 7C). Furthermore, continued evolution of SCNAs between subpopulations would also alter the SSNV divergence. For example, the SSNVs divergence is particularly enlarged for regions showing different CN states between the two samples (Fig. 7B). As such, increased SSNV diversity in regions with CN gains provides more materials for further selection within the expanding cell populations.

## Discussion

Despite the well-established link between a chaotic genome in tumors with the CIN phenotype and poor clinical outcomes[6], the mechanisms by which specific aberrations contribute to tumor growth remains poorly understood[38]. In this study, we have created a computational framework for measuring the arrival and initiation time of SCNAs during the somatic evolution of the MRCA of tumor sample(s), including complex gains with high CN states. By applying this method on multi-sampled WGS data of patient tumors, we have found that late truncal CN gains close to the most recent clonal expansion leading to the tumor sample(s) are common across multiple tumor types. Mathematical modeling predicts that these late evolving gains could contain final-expansion driver events, promoting the tumor growth. As CN gains increase the gene dosage and early somatic variants, we further demonstrated that incorporating the SCNA timing into an integrated genomic analysis has a strong potential for isolating the functional effect of specific aberrations.

Early genomic changes are presumably beneficial for tumor initiation[28], but it is unclear the effect of late truncal events. Here we have provided evidence that gains occurring late in the somatic evolution, i.e., close to MRCA, can also be beneficial. The simplified two-event cancer development model posits that the cancer-initiating event is followed by the promoting event[56]. We reason that the evolving CN gains might render the progenitor cell capable of "self-promoting," as they act similarly as a tumor "promoter" by (a) increasing the dosage of genes causing sustained proliferative signaling, (b) amplifying the mutant allele with early initiating driver variants, and (c) accelerating the accumulation of further genomic alterations. As both the early and late CN alterations could confer fitness advantages, chromosomal regions with SCNAs initiated early and arrived late, i.e., showing repetitive gains accompanying the entire course of the somatic evolution, could function as copy number "addictions." These underscore the value of `Butte` in identifying complex SCNAs with early initiation and late arrival.

GD, a landmark event in CN evolution, has a context-dependent fitness effect. The punctuated CN gains successfully induced the SRE in tumors that underwent a late GD. By contrast, for many other tumors, especially osteosarcoma, GD was followed by additional CN gains that produces the MRCA. GD could tolerate the occurrence of deleterious passengers[57]. However, simply escaping purifying selection was not sufficient to drive the ultimate outgrowth, at least not in the tumors with post-GD gains, where some chromosomal regions can reach higher CN states. Alternatively, GD may create an inflated genome space, accelerating the accumulation of driver alterations (Supplementary Fig. S13). Prolonged evolution following GD or arm-level aneuploidy before the MRCA has recently been associated with a poor prognosis in neuroblastoma[58]. As GD itself affects many genes, regions with pre- and/or post-GD gains could serve as a reduced search space for CN drivers.

Our method is applicable to a wide range of SCNAs, yet it is still challenging to analyze exceptionally high CN states (i.e., above eight). We note that regions with such a high CN likely evolve over time, such as the unequal segregation of extra-chromosomal oncogene amplifications[59,60]. As such, late changes are expected for these amplified regions. Some focal high-level gains could involve small segments where the number of SSNVs is inadequate for calculation. This problem can be mitigated by borrowing information from nearby segments with the identical CN state. This strategy is applicable to synchronized SCNAs, such as chromothripsis[61,62]. In addition, our analysis may have missed some late-appearing SCNAs due to overestimation of $t_K$ (Supplementary Figs. S1, S3). Based on benchmarking simulations and the CIs observed in real data, we recommend utilizing > 20 SSNVs for a SCNA segment, coupled with a read coverage depth > 50 and tumor purity > 0.5, to achieve a robust timing inference. Furthermore, deletion was not modeled as it is unidentifiable[18]; by comparing CN profiles between subpopulations, however, it is possible to study deletions during tumor expansion. Lastly, our framework relies on a constant SSNV rate per base per unit time within a region under SCNA evolution. We focused on the timeline of clonal SSNVs only as mutational signatures can differ in activity between clonal and subclonal lineages[63]. Specialized pre-selection of SSNVs that faithfully possess a clock-like behavior[19] is necessary for tumors whose mutagenic processes varies drastically within the studied timeline.

In our initial endeavor to understand the evolving gains in tumor development, our mathematical modeling focused on the timeline from germline to the tumor founder (or at least the founder of the dominant clone of the tumor). Our results suggest that if the MRCA forms a dominant clone of the tumor, the presence of abundant post-GD gains suggests that late gains may contain final-expansion drivers. Readers should note that even though the timing method works for any samples, the claim that late gains may contain final-expansion drivers relies on the condition that the MRCA of the sample is the founder of the dominant clone. In the analysis of individual samples, certain SCNAs may be associated with a subclone of considerable size, causing them to be identified as clonal. Consequently, some late gains could originate from subclonal events, although their impact on fitness has not been thoroughly investigated yet. Thus, we advocate the use of multi-region WGS data for analyzing late evolving gains.

Our findings also illustrate the existence of a fundamental connection between CN evolution and SSNV diversity, which can explain the positive correlation between aneuploidy and mutational burden when excluding hypermutated tumors[64,65]. Such a connection also indicates that we need to account for the dynamic nature of ongoing SCNAs when measuring subclonal evolution, which remains a challenge[66]. Finally, our results suggest that much can be gained by including the SCNA arrival time in studying tumor evolution, thereby shifting focus on exclusively early drivers to the evolving genomic events that affect the rate of tumor progression.

## Methods

### Somatic variant calling from WGS data

Raw WGS data in bam or fastq formats were downloaded from public databases provided by the original publications (Table 1) through the utilization of specific tools: pyega3 (version 3.4.0) for the European Genome-phenome Archive, sftp (version 3) for the Japanese Genotype-phenotype Archive, and sratoolkit (version 2.10.8) for the Database of Genotypes and Phenotypes. The cumulative read depth distribution along the human genome (hg38) and the tumor purity and ploidy for each sample are illustrated in Supplementary Figs. S23, S24, and S25. We have extended our existing pipeline, which had achieved a balance in sensitivity and specificity in detecting SSNVs by borrowing information across multiple samples[67,68], to allow the detection of clonal SCNAs and the breakpoints of structural variants.

SSNVs and INDELs: Analysis-ready read alignment bam files (against hg38) were generated according to the best practices, including indel realignment, base recalibration and flagging of duplicated reads. Raw candidate variants were produced by MuTect (v1.1.7)[3]. To reduce the false-positive rate due to misalignments or other technical artifacts and to salvage the variants that may be missed due to uneven read coverage between samples, the alignment features surrounding each candidate variant were collected for each sample. The heuristic-based criterion for the read alignment patterns was adopted to refine and variant calls[68]. Small insertions and deletions were called by using Strelka (v1.0.15)[69].

SCNAs: Copy number and tumor purity were estimated by using TitanCNA (v1.26.0)[25]. Germline heterozygous SNVs used as input to TitanCNA were identified using Samtools (v1.5)[70] and subject to the same filtering strategy as was applied to SSNVs. The one-clone solution reported by TitanCNA (i.e., the sample is dominated by a clone with an SCNA profile along the genome) globally fit the data of the read coverage and allelic imbalance well, with a few exceptions for which the two-clones solution are necessary to explain the data of specific genomic regions. The ploidy baseline (CN = 2) is determined by the model complexity in explaining the log read ratio and allelic imbalance of heterozygous SNPs (see Supplementary Fig. S26 as an example).

SVs: We incorporated two distinct SV calling tools relying on orthogonal approaches, i.e., DELLY (v0.7.8, abnormal read pair and split-read analysis)[71] and GRIDSS (v2.10.1, local assembly based algorithm)[72]. We focused on the SV breakpoints found by both tools, as these shared calls generally have higher quality (e.g., with higher breakpoint confidence) than those unique to each tool (Supplementary Fig. S27). SV breakpoints were annotated with AnnotSV[73].

### Analysis of genomic divergence

SCNA divergence: When multi-samples are available for a patient, the truncal and private SCNAs were identified as follows: (1) we partitioned the genome into disjoint segments by considering all the SCNAs called

from the samples of the patient; (2) for each segment, we calculated a generalized likelihood ratio statistics for the comparison between two samples. The statistics is the ratio of the values of the likelihood function (the probability of observing the read depth ratio and B-allele frequency for SNPs in the region) evaluated at the maximum likelihood estimation in the sub-model (two samples have the same CN profile) and at the maximum likelihood estimation in the full-model; and (3) the statistics converges weakly to a random variable with chi-square distribution and thus can be used to determine if a segment shows significantly different SCNA states between the two samples. The term "truncal SCNAs" refers to SCNAs that exhibit no difference in pairwise comparisons.

Sample phylogeny: We applied Treeomics (v1.7.13)[74] to construct sample phylogenies from SSNVs that are clonal in at least one specimen. Treeomics takes into account the uncertainty due to purity differences and variations of read depth on the SSNV loci to derive robust sample phylogenies. We note that Treeomics assumes sample homogeneity.

Clonality, multiplicity of SSNVs and SV breakpoints: SSNVs were categorized as either public (present in all tumor cells) or private based on their sharing patterns and allele frequencies in multi-sampling data[68]. In individual samples, clonal SSNVs were identified as those with the 95% confidence interval of cancer cell fraction (CCF) covering 1. We focused on the public SSNVs (multi-sampling) and clonal SSNVs (single sampling) for the timing analysis. For SSNVs or SV breakpoints existing in an SCNA region, we applied a binomial model to calculate the maximum likelihood estimates of the number of segment copies containing that variant[21].

**Allele state distribution of SSNVs in an SCNA.** For SSNV $i$ in an SCNA region (with CN configuration of $N_t : N_b$ and $M \geq 10$ SSNVs in total), we obtained from WGS the read counts carrying the mutant allele $m_i$ out of the total number of reads $d_i$. Expectation Maximization algorithm was used to estimate the proportion of SSNVs at each possible allele fraction, i.e., a vector $q$ that gives the probability of randomly acquired SSNVs in this region having a purity-adjusted allele frequency ($f_i = a_j$) for each possible allele state $\frac{j}{N_t}$. Note that we used the same symbol $a$ for allele frequency, as it is intrinsically tied to the allele state. To calculate the likelihood function of observing the particular SSNV data in an SCNA region, we sum across all possible allele states the product of $q_j$ and the probability that SSNV $i$ at $a_j$ has the observed read counts. Conditioned on the successful detection of the SSNV, the log-likelihood is given by,

$$\sum_{i=1}^{M} \log \Pr(m_i | m_i > 0, q) = \sum_{i=1}^{M} \log \left( \frac{\sum_{j=1}^{N_t - N_b} \Pr(m_i | f_i = a_j) q_j}{1 - \sum_{j=1}^{N_t - N_b} (1 - a_j)^{d_i} q_j} \right). \quad (3)$$

**Estimating the upper bounds of SCNA timing**

Assume that a clonal SCNA region (with $Nt : Nb$ at 5:1, Fig. 1D) has total $M$ SSNVs. We disregard deletions and model the CN evolution as 3 gain events, creating 4 stages of increasing CN states during the somatic evolution timeline from the germline to the founder cell of the tumor. Denote by the vector $\mathbf{t} = (t_0, \ldots, t_3)$ the fraction of time in each stage. We refer to $t_0$ as the initiation time, and $t_3$ (or $t_K$ where $K$ refers to the last stage) the lead time. Let $a_j$ represent a possible allele state for a SSNV, i.e., the fraction of the allelic copies with the SSNV. The possible values of $a_j$ in this case are $\{1/5, \ldots, 4/5\}$. Let the vector $\mathbf{q} = \{q_1, \ldots, q_4\}$ represent the fraction of the $M$ SSNVs having allele state for each of the possible $a_j$. Let $\mathbf{A}$ be a history matrix, with the entry $A_{jk}$ representing the number of copies in stage $k$ that would result in an allele state of $j/5$. In other words, all SSNVs originating from stage $k$ on those $A_{jk}$ copies will lead to the same allele state $j/5$.

Multiple paths or ordering of gain events could lead to the same SCNA state. In Fig. 1D, we graphically demonstrate the two possible histories of SCNA at 5:1. Our objective is to construct estimators of $t_0$ and $t_3$ using $\mathbf{q}$ and $\mathbf{A}$. Denote by $c$ the sum of the fraction of time multiplied by the number of copies in each stage. We have

$$c = 2t_0 + 3t_1 + 4t_2 + 5t_3.$$

Because the probability of a mutation occurring in stage $k$ is proportional to $t_{k-1}(k+1)$ (the fraction of the lifespan spent in stage $k$ multiplied by the number of copies in stage $k$), we have

$$q_j \approx \sum_{i=0}^{3} \frac{t_i A_{j(i+1)}}{c}. \quad (4)$$

We can write equation (4) in matrix form:

$$\mathbf{q} \approx \mathbf{A}\mathbf{t}/c, \quad (5)$$

When the history matrix $\mathbf{A}$ is invertible, we have

$$\mathbf{t} \approx c\mathbf{A}^{-1}\mathbf{q}, \quad (6)$$

and thus the desired estimators can be obtained. Note that in Fig. 1D, history $\mathbf{A}(\mathbf{a})$ induces an invertible history matrix, while the history matrix for history $\mathbf{A}(\mathbf{b})$ is singular. Therefore, the method introduced in ref. 18 cannot be directly applied.

For single and double gains ($Nt : Nb$ at 3:1, 2:0, 4:1 or 3:0), $t_0$ and $t_K$ are directly solved because matrix $\mathbf{A}$ is unique and invertible. For other SCNA states, `Butte` uses linear programming to obtain the upper bounds of timings across all possible history matrices for the corresponding CN configuration (Supplementary Fig. S28). Let $\mathbf{s}$ denote the vector of the column sum of matrix $\mathbf{A}$. Let $\mathbf{a}$ denote the vector of possible allele states. Abusing the notation, we now interpret $\mathbf{q}$ as the vector with entry $q_j$ representing the probability of a randomly acquired SSNV having allele frequency for each possible allele state $a_j$. Then the relation between $\mathbf{A}$ and $\mathbf{t}$ can be expressed as

$$\mathbf{A}\mathbf{t}/(\mathbf{s}^T \mathbf{t}) = \mathbf{q}. \quad (7)$$

From equation (7), we have

$$(\mathbf{A} - \mathbf{q}\mathbf{s}^T)\mathbf{t} = \mathbf{0}, \quad (8)$$

where $\mathbf{0}$ represents a vector with all 0's. Since $\mathbf{t}$ denotes the time fractions of different CN evolution stages, we have

$$\mathbf{1} \cdot \mathbf{t} = 1, \quad (9)$$

where $\mathbf{1}$ represents a vector with all 1's. `Butte` solves the following optimization problem by linear programming:

$$\begin{aligned} \max_{\mathbf{t}} \quad & t_K \\ \text{s.t.} \quad & (\mathbf{A} - \mathbf{q}\mathbf{s}^T)\mathbf{t} = \mathbf{0} \\ & \mathbf{1} \cdot \mathbf{t} = 1, \end{aligned}$$

where $t_K$ is the last element in vector $\mathbf{t}$. The maximum value of $t_K$ gives us an upper bound of the lead time given $\mathbf{A}$. For upper bounds of initiation times, we instead maximize $t_0$ which is the first element in $\mathbf{t}$. To tolerate noise in the allele state distribution estimated from sequencing data, we add a slack variable on each capacity constraint, having a penalty cost of 100. The confidence intervals of the estimated upper bounds were calculated through bootstrapping the SSNV data.

## Benchmarking the timing method

To evaluate the robustness of Butte, we simulated SSNV data for a SCNA region across different evolutionary histories, depth of coverage ($D$), number of available mutations ($M$) and tumor purity ($P$). Given a known evolutionary history of a SCNA and known timing for each time stage, we generated the allele state distribution vector **q**. Given the total number of SSNVs $M$ available for the SCNA region, the number of SSNVs at distinct allele states were drawn from a multinomial distribution with probability for these allele states (**q**) for each iteration. Given the tumor purity $P$, the true allele frequency $f$ of a mutation at a known allele state was calculated. The sequencing depth of mutations ($D$) follows a negative binomial distribution $D \sim \mathbf{NB}(u_D, \sigma_D)$, with mean $u_D$, and dispersion $\sigma_D = u_D/10$. The resulting read counts of the mutant allele having a true allele frequency of $f$ follows a binomial distribution $m \sim \mathbf{Bin}(D, f)$.

To assess the precision, true positive rate (TPR), and false positive rate (FPR) of both early and late event detection, we conducted simulations for SCNA state ($Nt$:$Nb$) at 6:2 and 6:1 Supplementary Figs. S11–S12. To evaluate late gains, we use a threshold value ($T$) and set the initiation time ($t_0$) at 0.1, 0.2, and 0.3, respectively. Subsequently, we simulated SSNV data using randomly generated lead time $t_K$ values. True positives were identified when the predicted $t_K$ was less than $T$ and the actual $t'_K$ was also less than $T$. False positives occurred when the predicted $t_K$ was less than $T$ but the actual $t'_K$ exceeded $T$. Conversely, true negatives were cases where both the predicted and actual $t'_K$ values were greater than $T$, and false negatives were instances where the predicted $t_K$ exceeded $T$ but the actual $t'_K$ was less than $T$. Similar criteria were applied for early gains by comparing the predicted and actual $t_0$ values with the threshold ($T$). The performance metrics were evaluated across different threshold values ($T$).

To examine tumors from[29] using Butte (Supplementary Fig. S8), we acquired SSNV and SCNA predictions for the same samples from the PCAWG (PanCancer Analysis of Whole Genomes) dataset[26] via the ICGC data portal. Subsequently, Butte was applied to the downloaded dataset, comprising 15 tumor samples with event timing predictions previously reported by ref. 29. These predictions included events such as tetraploidy, trisomy, tandem duplication, and GD, determined using the graph theory-based method[21].

## Determining the timing of genome doubling

We identified clustered gains by clustering the inferred timing via non-parametric density estimation (R-package pdfCluster)[75]. We define GD as the prominent and concentrated burst of gains, containing more than 40% of all timed segments. A cutoff of 40% seems suitable for identifying the prominent burst of gains, as depicted in Supplementary Fig. S29. It's important to emphasize that we do not assume duplications necessarily cover the entire genome during GD events. Gains in these clusters have similar timing estimates (standard deviation $\sigma \sim 0.1$) (Supplementary Fig. S30), suggesting that they occurred within a narrow time window. We regard the averaged timing of all the segments in the corresponding cluster as the timing of GD. Post- and pre-GD events were identified as those occurred $\pm 1.3\sigma$ away from the GD, respectively.

## Detecting recurrent early or late gains

We partitioned the genome into bins of 1 million base pairs each and ranked these bins in each sample based on their respective timing values ($t_0$ for BRCA and $1-t_K$ for OS, respectively). To avoid ties, we introduced jitter to the timing values. Subsequently, we calculated the deviation from the middle rank of each sample for each bin. This middle rank represented the expected value under the null hypothesis, signifying no recurrent early (for BRCA) or late (for OS) gain regions across patients. For each tumor type, we aggregated these rank deviations across patients for each bin. Normalizing these rank sums by their standard deviations produced standardized values, which approximately followed a standard normal distribution if the null hypothesis held true. A significantly negative standardized rank sum

indicated recurrent early initiating gains, while a markedly positive value indicated recurrent late establishment gains. Simultaneously, to assess the prevalence of gains in each genomic bin across patients, we ranked the segment mean (which represented the read depth ratio between tumor and normal samples) in a similar manner as the timing values. Subsequently, we applied the same rank sum normalization technique. A notably positive normalized rank sum for the segment mean would indicate frequent high copy number gains across patients.

## Functionality of genes affected by late gains

To see which cancer hallmarks are associated with late gains, we performed Gene Set Enrichment Analysis (GSEA)[76] on the gene list ranked by the averaged arrival time across patients of SCNAs affecting a corresponding gene. We used the gene sets representing hallmarks of cancer[1] from COSMIC database[77]. R-package fgsea[78] was utilized to perform the GSEA analysis with 50000 permutation.

To further evaluate the fitness effect of genes affected by late gains, we analyzed the gene Chronos score[50] (gene knockout fitness effect) provided in the DepMap database. The Chronos score reflects the change in cell proliferation upon the CRISPR knockout of the respective gene in a particular cell line. A lower negative Chronos score indicates that the gene is a denpendency in a cell line. For simplicity, we took the average score for each gene across all the cell lines. For each tumor, we calculated the fraction of genes affected by late gains with a mean Chronos score $< -0.5$. We then obtained the normalized ratio (NR) by dividing it to the ratio calculated from all the genes in the database. To get a background of the NR of randomized genomic regions, given a set of segments showing late gains, we randomly sampled regions by keeping the same segment lengths by using R-package regioneR[79]. We then compared the normalized ratio between patient data and the randomization (Supplementary Fig. S22).

## Mathematical modeling of late evolving gains

Consider two contrasting models based on multi-type branching processes with mutations. In both models, the tumor grows from a single tumor-initiating cell which just acquired GD. During the tumor's progression, cancer cells accumulate mutations (post-GD gains). In the base model (neutral model), all mutations are passenger mutations. Therefore, all cancer cells give birth at a rate of $a_0$ and die at a rate of $b_0$. The net growth rate is $\lambda_0 = a_0 - b_0 > 0$. Neutral mutations occur at rate $u_0$ per unit time throughout the lifetime of a cell, and each mutation is distinct following the infinite-sites model[80]. Clonal post-GD gains are those acquired prior to the cell division (or the onset of expansion) leading to two surviving sublineages. Let this division event (denoted by an effective birth) occur at a rate of $\lambda_0$ (cf. page 10 of[42]) conditioned on the non-extinction of the population. By the memoryless property of the exponential distribution, counting the number of post-GD gains prior to the first effective birth is analogous to counting the number of tails until the first head in a sequence of coin tosses, where the probability of a head (effective birth) is $\frac{\lambda_0}{\lambda_0 + u_0}$ and the probability of a tail (neutral mutation) is $\frac{u_0}{\lambda_0 + u_0}$. Therefore, the number of gains before the first effective birth follows a geometric distribution with parameter $\frac{\lambda_0}{\lambda_0 + u_0}$ and mean $\frac{u_0}{\lambda_0}$. We then investigated the number of mutations which are shared by more than 90% of the total population (we refer to them as dominant mutations). Gunnarsson and his co-authors[43] derived exact expressions for the expected SFS of a cell population that evolves according to a branching process. We utilized their results on the skeleton subpopulation (see Appendix C of[43])−cells with an infinite line of descents which determines the high frequency spectrum−to express the expected number of dominant mutations $\bar{S}$ when the tumor reaches a fixed size $N$ as

$$\bar{S} = \frac{N}{\lceil 0.9N \rceil} \cdot \frac{u_0}{\lambda_0} \approx 1.11 \frac{u_0}{\lambda_0}. \tag{10}$$

In the alternative model (selection model), the tumor-initiating cell and its descendants with only passenger mutations form the type 0 population. Type 0 cells give birth at a rate of $a_0$ and die at a rate of $b_0$. The net growth rate is $\lambda_0 = a_0 - b_0 > 0$. Type 0 cells mutate to type 1 cells at a rate of $u_1$. Type 1 cells give birth at a rate of $a_1$ and die at a rate of $b_1$. The net growth rate is $\lambda_1 = a_1 - b_1 > \lambda_0$. Both type 0 and type 1 cells accumulate passenger mutations at a rate of $u_0$. We assume that when the tumor is sampled, the descendants of the first type 1 cell with infinite lineage dominates the population. This is usually the case in our simulations where the fitness advantage conferred by the bene-ficial post-GD gain is large. We note that due to stochasticity, descen-dants of a later-appearing type 1 cell (e.g., the second or the third one, etc.) can also dominate the population. However, type 1 cells acquired later would accrue more post-GD gains on average and thus our claim stays valid. On the other hand, it is the equivalent of the base model if no type 1 cells dominates the population upon sampling. In Lemma 1, we obtained the distribution of the time to the first type 1 cell with infinite lineage conditioned on the non-extinction of the tumor.

**Lemma 1.** Let $\sigma_1$ denote the time of occurrence of the first type 1 cell that gives rise to a family which does not die out, and let $\Omega_\infty$ denote the event of non-extinction of the tumor. Then

$$\mathbb{P}(\sigma_1 > t | \Omega_\infty) = \frac{a_0(1-q_0) + \frac{u_1(1-q_1)}{1-q_0}}{a_0(1-q_0) + \frac{u_1(1-q_1)}{1-q_0}e^{\zeta t}},$$

where

$$q_0 = \frac{a_0 + b_0 + u_1 - \sqrt{(a_0+b_0+u_1)^2 - 4a_0(u_1q_1+b_0)}}{2a_0},$$

$$q_1 = \frac{b_1}{a_1}, \quad \text{and}$$

$$\zeta = \frac{u_1(1-q_1)}{1-q_0} + a_0(1-q_0).$$

With Lemma 1, we can obtain the expected number of passenger mutations accumulated in the first type 1 cell with infinite lineage, denoted by $\bar{S}$:

$$\bar{S} = \int_0^\infty \mathbb{P}(\sigma_1 > t | \Omega_\infty) u_0 dt. \tag{11}$$

With (10) and (11), we can obtain that the expected number of domi-nant post-GD gains in the subpopulation generated from the first type 1 cell with infinite lineage is $1.11\frac{u_0}{\lambda_1} + \bar{S} + 1$, where the last 1 represents the number of post-GD driver gains. Proof for Lemma 1 and details of (10) can be found in Supplementary Methods.

**Mathematical modeling in the context of multiple drivers.** In our multi-type branching process model, we examine two driver muta-tions: mutation one and mutation two. We initiate with a single cell devoid of mutations, which possesses a birth rate of $a_0$ and a death rate of $b_0$. Mutation one, acquired at a rate of $u_1$, adds $\delta_1$ to the birth rate ($a_1 = a_0 + \delta_1$). Mutation two, acquired at a rate of $u_2$, adds $\delta_2$ to the birth rate ($a_2 = a_0 + \delta_2$). Cells with both mutations have a birth rate of $a_3 = a_0 + \delta_1 + \delta_2$. Death rate remain the same as $b_0$. We simulated tumor growth from a mutation-free cell until the first cell acquired both dri-vers without going extinct (100,000 for each parameter set). We cal-culated the fraction of cases where mutation one occurred first.

**Reporting summary**
Further information on research design is available in the Nature Portfolio Reporting Summary linked to this article.

## Data availability

The article, its Supplementary Information files, and source data files contain all pertinent data supporting the main findings of this study. The raw WGS data analyzed in this paper, previously published, are sourced from the following datasets: EGAD00001004482, EGAS00001000263, EGAD00001004966, phs001722.v1.p1, EGAD00001002696, JGAD000095, EGAD00001000891, EGAD00001001394 and are detailed in Table 1. The raw data are subject to controlled access in accordance with the specific data sharing policies mandated by each data provider. Access can be acquired by submitting a request to the respective data access committees and adhering to their spe-cified sharing policies. Instructions for requesting access are provided on the respective databases. The processed timing result are available at https://sunpathlab.github.io/Datasets/. The datasets containing Chronos scores or CRISPR gene effects in the DepMap database can be downloaded by visiting https://depmap.org/portal/download/all/(ver-sion 22Q2). The dataset that are necessary to interpret, verify and extend the research in the article are provided in the Supplementary Information and Source Data file. Source data are provided with this paper.

## Code availability

All the original code for `Butte` (a computational framework for esti-mating SCNA arrival and initiation time from WGS data) and the associated mathematical modeling have been deposited in a GitHub repository, publicly accessible through https://github.com/SunPathLab/Butte/. The released version utilized in this paper is accessible on Zenodo[81]. Code for whole genome sequencing analysis can be found in package `ith.Variant` through https://github.com/SunPathLab/ith.Variant.

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

## Acknowledgements

Research in the Sun Lab is supported by NIH grant R01CA276666. We thank the Rein in Sarcoma foundation for supporting the early preliminary work of this study. This study uses the computing resources in Minnesota Supercomputing Institute. This manuscript was prepared using limited access datasets obtained from BC CANCER (OS_N), Queen Mary University of London (COAD_C), the Cancer Genome Project in Wellcome Sanger Institute (BRCA_Y), and does not necessarily reflect the opinions or views of the corresponding provider institutions. The OS_C dataset was generated by the St. Jude Children's Research Hospital - Washington University Pediatric Cancer Genome Project. The BRCA_K dataset was originally generated by research led by Dr. Masahito Kawazu and available at the website of the National Bioscience Database Center (NBDC; http://biosciencedbc.jp/en/) of the Japan Science and Technology Agency (JST). We thank International Cancer Genome Consortium (ICGC) for providing the access to the PRAD_G dataset (originally produced by Cancer Research UK Prostate Cancer Group Study) and ESCA_R dataset (by the Oesophageal Cancer Clinical and Molecular Stratification Study Group). We would also like to acknowledge the database of Genotypes and Phenotypes (dbGaP) and Dr. Ryan C. Fields for producing the COAD_D dataset (phs001722.v1.p1) which was supported by Siteman Cancer Center Investment Program. We thank Dr. Boyang Liu for commenting on the manuscript. We thank Dr. Gunnarsson for sharing his proof ideas on the skeleton of the branching process.

## Author contributions

R.S. and Z.W. designed the study. R.S., Y.X. and Z.W. developed the algorithms. Z.W., R.S. and A.N.N. constructed mathematical models and performed simulation studies. R.S., Y.X., Z.W. and L.M. performed the analysis of WGS data and visualized the results. R.S., Z.W., Y.X., N.M., S.M.D and J.M.S interpreted the results and wrote the manuscript. All authors reviewed and provided feedback on the manuscript.

## Competing interests

The authors declare no competing interests.
