## [Peer Review File · Nature Communications]

REVIEWER COMMENTS

Reviewer #1 (Remarks to the Author): Expert in cancer genomics and evolution, and mathematical modelling

Wang and colleagues develop a method called “Butte” to estimate the timing (initiation and arrival) of somatic copy number gains from whole genome sequencing data. Their approach advances the current state of the field on SCNA timing by being able to time high copy number gains ($CN > 4$), up to a copy number of 8. Multiple trajectories are possible to generate some CN states, and their linear programming approach accounts for this by giving the upper bound on the timing of events. They perform extensive validation on simulated datasets, and then apply Butte to eight publicly available datasets. They also employ a branching process model to investigate late gains.

I believe this work provides a valuable contribution to the field. In my assessment, the methods and models used are sound, and I appreciate the significant validation performed and reported in the supplemental figures. They demonstrate the general utility of their method by applying it to multiple cancer types spanning 8 different datasets, and provide meaningful analyses of these tumors. One of my main concerns relates to the bias in the estimates, especially when drawing conclusions about late events from the timing estimates that are overestimated. I would like to see some additional validation performed about the early and late classifications. Some revision of the logic flow and precision of terminology would improve the readability and presentation of results. Lastly, I have some concerns about the conclusion from the mathematical modeling work that late gains may contain rate-limiting drivers. I have listed my specific comments below, separated into major and minor issues.

Major

1. Because the estimates are upper bounds for some CN profiles and tK estimates are often overestimated, I'm concerned about overinterpreting the significance of “late” gains, which are defined based on these biased estimates. Figures S1-S3 show tK estimates are usually overestimated-- significantly so for smaller true values of tK. Can the authors comment on how often it will be possible to accurately identify early events? I think a similar validation analysis to Figures S2-S5 assessing the true positive rate of early predictions and false positive rate of late gains would clarify how much can be interpreted from the assignment of late and early gains.

2. Why does Butte underestimate t_0 , even though the estimate is an upper bound? Does Butte overestimate t_K , even when the estimate is not an upper bound (exact solved timing)? What is causing these biases?

3. What about other possible trajectories to the observed CN profile, involving more complicated structural variants/SCNAs? How will it affect timing estimates? For example, within a region of elevated CN, is CN always constant? If not, how are different possible trajectories to that CN profile accounted for? E.g. What if there's a continuous region with copy number 3-4-3?

4. Overall clarity of terminology and results. Clarity of the manuscript could be improved, especially with regards to usage of the terms "early" and "late" gains, as well as differentiating between GD and pre- and post-GD gains. Some specifics:

a. Once methodology is introduced (in the results), can you more explicitly define early and late gains? In the introduction it says late gains are those occurring in last 20% of evolution time and early as occurring in the first 20%. It is unclear how these definitions are applied in practice in the results section. Can the definition be related to the t_0 and t_K estimates, since it presumably depends on them?

b. Figure 4 is unclear about when the GD events are occurring. In the corresponding parts of the text, the time of GD events in Figure 4 are mentioned. But as far as I can tell, Figure 4 shows all SCNA gains, including non-GD ones (since pre- and post-GD gains are discussed). Could annotations to the figure or clarifications to the text be provided? This is especially important to understanding the results, since there is much discussion about gains that occurred before and after GD. For example, could vertical lines indicate the times for GD and early and late cutoffs, similar to how GD time is shown in Fig S21.

5. Other Figure 4 issues.

a. It's difficult to see the difference between initiation and arrival in plots like 4A. With the number of events plotted, it is difficult to see which events are connected by the dotted lines that link initiation and arrival. Currently they look like two separate events.

b. Fig 4A left panel for CN is ambiguous with current color choices. I believe right $\sim 1/3$ of this panel is showing any LOH, but this color is similar to CN = 0. Can LOH color be more obvious difference from those used to indicate CN = 0? Or LOH can be indicated in a different way from CN (e.g. box around affected region).

6. I am not convinced by the conclusion that late gains may contain rate-limiting drivers based on the modeling work. In particular, the mathematical modeling section concludes with the strong statements in lines 301-304, which are not adequately supported by the modeling results. I think there is too much causation implied in these statements, and the logic doesn't flow. This conclusion is restated throughout (e.g., discussion, abstract, etc.). Even if I'm misunderstanding this conclusion, I think it should at least be rephrased or explained further.

7. Some confidence intervals for timing estimates are very large. Can the authors comment on the magnitude of the uncertainty for the timing estimates? For the tumors analyzed, can they provide summary statistics for the size of the confidence intervals (in units of evolution time spanned).

Minor

1. Running title has hyperlink to unrelated website.
2. Define SRE in the introduction at its first mention.
3. Line 543. When it is stated in the Methods how to determine GD, it is unclear to me why after clustering inferred times, GD would correspond to the cluster containing 40% of segments. Is this a standard way of identifying GD? If not, how does this compare to other methods for identifying GD?
4. What is the "evolution time" mentioned in the intro? This term should be defined.
5. Why do estimates sometimes have increasing error with increasing read depth (e.g. Fig S3)?
6. Is there not difficulty in resolving number of SSNVs at j/N frequencies as copy number N increases?
7. Top left panel on Fig 2 could use more annotation or explanation.
8. I think the recent Nature Genetics paper by Körber et al (<https://doi.org/10.1038/s41588-023-01332-y>) is relevant to the conclusions and methods discussed.
9. Line 193: Clarify that "the existing method" is from Cross et al. 2018.
10. Line 235: "The earlier the timing of GD, the more post-GD CN gains (Fig. 4A)." Fig 4A shows 3 example tumors, so this statement should either be revised to reflect this fact, or reference a figure more appropriate for the generality of this statement.
11. Can equation 2 at line 287 (or equivalently in Methods, Equation 11 (line 613)) be explained and/or derived?
12. Line 273: "Moreover, numerical simulations show that the number of dominant post-GD gains continues to follow a geometrical-like distribution with the mode at zero." Can the authors support this statement with a reference to a figure?

13. Can the authors include citation(s) for the strong and general opening sentence of the discussion (line 350)?

14. Lines 399-401: "Specialized pre-selection of SSNVs that faithfully possess a clock-like behavior (Gerstung et al.) is necessary for tumors whose mutagenic processes varies drastically within the studied timeline." Is that not the case for the tumors studied?

15. Line 48: "The inextricable relation of SCNAs to cancer initiation (Nowak et al. ; Davoli, Xu, et al) and progression (Lukow et al.) has become a consensus in cancer genomics." I'd be hesitant to use the word "consensus" here, even though it is a well-observed relation. It implies complete agreement in the literature about the primacy of SCNAs in cancer initiation and progression, excluding other important factors like microenvironment, smaller genetic alterations, epigenetics, etc. I don't think there is much doubt that SCNAs play an important role, but I think this phrasing might imply more than is intended.

16. The timing ranks in Fig. 6A needs more explanation or visual cues.

Reviewer #2 (Remarks to the Author): Expert in cancer genomics and evolution, mathematical modelling, and bioinformatics

In this paper, the authors present an extension of an existing algorithm by Purdom et al. (2013) to predict the timing of copy number gains in cancer tumors, including repeated events at the same locus up to a copy number of 6. This algorithm is applied to several cancer DNA sequencing datasets. For each genomic locus affected by SCNAs, the timings of the first gain and the last gain before population expansion of the sample are estimated. The derived results can provide insight into the dynamics of CN gains in human cancers. However, the progress over existing approaches and findings as well as the validity of the interpretations of the main results remain somewhat uncertain.

Major comments

1. First, the authors should choose what they mean by "arrival time" and maintain this throughout the manuscript. In some places, it is the duration t_K of the interval after the last CN gain before expansion (e.g. caption of Fig. 1), whereas in others it is the time at which the last CN gain takes place, measured relative to birth (e.g. Fig. 1A, Fig. 4), i.e. $1-t_K$.

2. The estimation of t_K is based on the VAF spectrum. In general, the VAF spectrum contains a number of low-frequency variants that occur in the subclonal phase of tumor growth. If present, these could inflate the estimates of t_K . According to the Methods, these variants are filtered out by restricting to clonal variants. However, since the definition of clonal vs subclonal already relies on knowledge of copy number status and sequence of mutational events (to calculate the CCF, l. 466), how could clonality status in the single-sample tumors be reliably inferred?

3. Another putative confounding factor for the proposed inference is sample purity. Since variants with an alternate read count of less than 3 are usually filtered out of the sequencing data, low purity of a sample means that certain variants will never be seen, even after correcting for purity by dividing the VAF. This would result in a downward bias of t_K . The authors evaluated the effects of purity on the estimation of t_K (Fig. S2,3). In these simulations, was realistic filtering of variants with low alternate read count applied? Do the authors find a correlation between the estimated t_K and purity in the real data? And with the read depth and the number of SNVs?

4. One of the main results is the shift toward small t_K of some putative late driver events such as MYC for osteosarcoma in Fig. 4C, placing them close in time to the MRCA of the sample. However, are the small t_K s observed for such events not a necessary consequence of the high CN? In other words, the more events that must take place (i.e. the higher K), the smaller the fraction of time available (t_i) for each of them, since $\sum_i t_i = 1$. This would naturally decrease t_0 and increase $1-t_K$. I do not think that the differences in $1-t_K$ (and t_0) between regions, as shown in Fig. 4C, can be meaningfully interpreted without taking this intrinsic property of the data into account. Similarly, the results shown in Fig. S12 seem expected, since, for example, the 3:1 condition involves one fewer SCNA event than 3:0.

5. Regarding the analysis in Fig. 5, I am not sure I follow the conclusion from the mathematical model that late advantageous gains become a rate-limiting event (l. 304). First, it is not clear why Fig. 5A would not be consistent with neutral evolution, since the number of post-GD gains should be linearly proportional to the mutation rate and the time available for their occurrence, which the figure does show. In truncal evolution, neutral passenger SCNAs, like selected SCNAs, are dominant as soon as they occur. Finally, it appears that neutral evolution is rejected by assuming that u_0 and λ_0 are comparable (l. 272). It is unclear why this is assumed, given how little is known about cancer cell growth and mutation rates. The statement is weaker in the caption of Fig. 5 ("modeling suggests that advantageous CN gains can occur late") and in l. 356, but then it is tautological.

6. As I understand, each history matrix gives an estimate for each t_i . If a matrix is not invertible, the estimate of t_i is replaced by an upper bound for t_i (only applied to $i=0$ and $i=K$). How are these two qualitatively different quantities (i.e., exact estimates vs upper bounds) combined in diagrams such as Fig. 4B,C?

7. To benchmark the method, I would recommend comparing it to existing methods such as that of Greenman et al. (2011) by applying it to the data in that paper and that of Nik-Zainal (2012), which used the same algorithm, to compare the event time predictions.

8. In addition to applying the method to datasets with existing predictions of CN gain timings, the most compelling benchmark with the available data would be to apply the algorithm to subclones identified in the phylogenetic reconstructions of the multi-region sequencing datasets that carry private SNCAs (Fig. 3). Fig. S8 nicely shows that subclone-private gains in the COAD cohort are indeed predicted to occur later than shared gains on average. Why did the authors not perform the same analysis for all other cohorts?

9. As another way to show the added value of the method, compare the GSEA predictions from Fig. 6A with the ranking of genes simply by their average CN (as opposed to average $1-t_K$). If the enrichment scores are worse, this would be a nice demonstration of the utility of the method. Are p-values or adjusted p-values shown in Fig. 6A? Since several gene sets were tested, a correction for multiple testing needs to be applied.

Minor comments

10. The derivations of the relationship between SSNV frequencies and CN gains on which the algorithm is based assume that all gains are truncal, resulting in integer N_t values. However, SCNAs that arise (shortly) after expansion would be subclonal in the sample, but could produce similar patterns. In this case, the true history would not match any of the history matrices used. Since the algorithm assumes a fixed set of history matrices and integer N_t values, how will the results be affected if such subclonal gains are present at an SCNA site? For example, in the VAF spectrum shown in Fig. 2, the t_0 state is not at the expected frequency of $5/6=0.83$, but rather at 0.75.

11. In the lower panel of Fig. S2, a non-invertible history matrix is shown, but distributions of exact derived t_K values are given, although in this case only upper bounds can be derived. How can this be?

12. It is not quite clear to me what the authors want to say with Fig. 7. Since the mutation target increases with increasing CN, it is expected that the number of SSNVs correlates with CN. That these SSNVs may be subject to selection is also quite expected.

13. The use of the term genome doubling (GD) is confusing. First, there is no definition in the main text of what is the difference between GD and other CN gains. In the Methods, it appears that GD was derived from density-based temporal clustering of SCNAs, and whenever more than 40% were found in

one cluster, these SCNAs were considered part of a GD event. The goal of calculating GD appears to be to allow comparison with previous studies that have already assessed the relative rate of SCNA gains and losses before and after whole-GD (WGD). However, in these studies, GD is usually associated with a global, i.e. whole-genome, doubling of ploidy. Why was this definition of GD used rather than a more standard derivation of WGD?

14. The definition of A_{jk} (l. 153) seems unnecessarily complicated. It is the number of segment copies in stage k that produce the final allele state (i.e. frequency) $a_{j=N_t}$.

Authors' Response to Reviews of

Evolving copy number gains promote tumor expansion and bolster mutational diversification

Zicheng Wang, Yunong Xia, Lauren Mills, Athanasios N. Nikolakopoulos, Nicole Maeser, Scott M. Dehm, Jason M. Sheltzer and Ruping Sun

RC: *Reviewers' Comment*, **AR:** Authors' Response, Manuscript Text

General Comments

Note: Line numbers and Figure indexes refer to the revised manuscript version with changes being highlighted in **blue**. The figures shown in this response are indicated in black font color and include hyperlinks.

1. Response to Reviewer #1

1.1. General comment

RC: *Wang and colleagues develop a method called “Butte” to estimate the timing (initiation and arrival) of somatic copy number gains from whole genome sequencing data. Their approach advances the current state of the field on SCNA timing by being able to time high copy number gains ($CN > 4$), up to a copy number of 8. Multiple trajectories are possible to generate some CN states, and their linear programming approach accounts for this by giving the upper bound on the timing of events. They perform extensive validation on simulated datasets, and then apply Butte to eight publicly available datasets. They also employ a branching process model to investigate late gains.*

RC: *I believe this work provides a valuable contribution to the field. In my assessment, the methods and models used are sound, and I appreciate the significant validation performed and reported in the supplemental figures. They demonstrate the general utility of their method by applying it to multiple cancer types spanning 8 different datasets, and provide meaningful analyses of these tumors. One of my main concerns relates to the bias in the estimates, especially when drawing conclusions about late events from the timing estimates that are overestimated. I would like to see some additional validation performed about the early and late classifications. Some revision of the logic flow and precision of terminology would improve the readability and presentation of results. Lastly, I have some concerns about the conclusion from the mathematical modeling work that late gains may contain rate-limiting drivers. I have listed my specific comments below, separated into major and minor issues.*

AR: We appreciate your positive feedback on the paper and thank you for highlighting concerns related to bias in estimates, logic flow, precise terminology, and the clarity of mathematical model interpretations. In this revised version, we have made efforts to address all the points you raised.

Specifically, we have conducted a performance evaluation by assessing the precision, true and false positive rates for calling both early and late gains. Relevant sections have been expanded accordingly. To enhance precision, we have introduced the term ‘lead time’ to represent the period from a region’s last gain to the time of the Most Recent Common Ancestor (MRCA). Moreover, we have replaced the term ‘rate-limiting driver’ with ‘final-expansion driver’ to emphasize the role in the tumor’s ultimate expansion. The Mathematical Modeling section has been expanded and restructured for better clarity, emphasizing our conclusion that the ‘final-expansion driver’ likely occurs close to the MRCA. Additionally, we have included simulations to discuss the role of early gains.

Please consult our detailed responses to your specific comments for a comprehensive understanding of the revisions made.

1.2. Major concern

RC: *1. Because the estimates are upper bounds for some CN profiles and t_K estimates are often overestimated, I'm concerned about overinterpreting the significance of "late" gains, which are defined based on these biased estimates. Figures S1-S3 show t_K estimates are usually overestimated—significantly so for smaller true values of t_K . Can the authors comment on how often it will be possible to accurately identify early events? I think a similar validation analysis to Figures S2-S5 assessing the true positive rate of early predictions and false positive rate of late gains would clarify how much can be interpreted from the assignment of late and early gains.*

AR: Thank you for your valuable comments and suggestions. We would like to clarify that t_K represents the duration from the establishment of the observed copy number state (the last gain) to the onset of population expansion (the MRCA), and in the revised manuscript, we have defined it as the **lead time**. This is distinct from the **arrival time**, which signifies the duration from germline to the occurrence of the last gain. Overestimation of t_K results in the underestimation (or omission) of late gains, rather than an overcalling of them. Furthermore, the overestimation of t_K is noticeable only for specific copy number configurations, such as 6:2. Consequently, for these SCNA states, there is a risk of missing some late gains.

Per your suggestion, we investigated precision, true positive rate (TPR), and false positive rate (FPR) of both early and late events detection, where

$$\begin{aligned} \text{precision} &= \frac{\text{true positive}}{\text{true positive} + \text{false positive}} \\ \text{true positive rate} &= \frac{\text{true positive}}{\text{true positive} + \text{false negative}} \\ \text{false positive rate} &= \frac{\text{false positive}}{\text{false positive} + \text{true negative}}. \end{aligned}$$

To accomplish this, we established a predefined threshold (T) for timing. Regarding early gains, if the predicted t_0 is less than T , we classify true positives as cases with actual $t'_0 < T$ and false positives as cases with actual $t'_0 > T$. Conversely, if the predicted t_0 exceeds T , we categorize true negatives as cases with $t'_0 > T$ and false negatives as cases with $t'_0 < T$. These definitions are similarly applied to late gains by comparing T with the predicted t_K and actual t'_K . Performance is then evaluated with different T .

In a manner similar to our approach with Supplemental Figs. S2-S5, we conducted simulations for copy number configurations 6:2 and 6:1, each with two different history matrices. The results are illustrated in Supplemental Figs. S12-S13. Generally, the three metrics (Precision, TRP, and FPR) used to determine early or late gains exhibit an increase with T . Specifically, for $T = 0.2$ in configuration 6:2, the precision values are approximately 0.75-0.8 for t_K and 0.65-0.75 for t_0 . The FPR is less than 0.1 for t_K and below 0.2 for t_0 . The TPR is above 0.8 for t_0 but can be less than 0.5 for t_K . This aligns with our reasoning; the overestimation of t_K itself could result in the omission of late gains. In other words, it is a bit conservative in calling late gains for 6:2. The observed trends for these metrics in the case of 6:1 configurations are similar to those seen in 6:2 configurations. At the threshold $T = 0.2$, the false positive rate for t_K remains below 0.2, and the true positive rate for t_0 consistently exceeds 0.8 in our simulations.

RC: *2. Why does Butte underestimate t_0 , even though the estimate is an upper bound? Does Butte overestimate t_K , even when the estimate is not an upper bound (exact solved timing)? What is causing these biases?*

AR: We value the reviewer’s feedback and concerns. The underestimation of t_0 is primarily noticeable in specific copy number states, such as 6:2, as opposed to 6:1. Furthermore, this underestimation is more pronounced when the actual t_0 is relatively large (≥ 0.5), allowing limited time for intermediate stages (see Supplementary Fig. S5). We attribute this underestimation to (1) the penalty for infeasibility in our linear programming approach; and (2) the characteristics of the history matrices for the 6:2 copy number configuration is more sensitive to the estimated allele state distribution (q vector). See an example history matrix for 6:2,

$$\begin{bmatrix} 0 & 0 & 2 & 4 & 6 \\ 1 & 3 & 2 & 1 & 0 \\ 0 & 0 & 0 & 0 & 0 \\ 1 & 0 & 0 & 0 & 0 \end{bmatrix}.$$

The accuracy of timing estimates is closely linked to the estimated q vector, representing the proportion of SSNVs at each allele state. In the scenario of 6:2, t_0 contributes to SSNVs with allele states 4/6 and 2/6. Estimation errors on the q vector is likely to make the true history infeasible and this effect is more pronounced when the fraction of intermediate stages is small (i.e., t_0 is relatively large). This is because the errors in the estimates of SSNVs at 4/6 and 2/6 often occur in opposite directions. Since Butte currently places a large penalty on infeasibility, the estimation of t_0 from the true history (which is indeed

an upper bound of t_0 for the given q vector) might not be selected (i.e., `Butte` prioritizes solutions from the history matrix that result in feasible models over those leading to infeasible models, although the true history might be among the latter ones) and t_0 can be underestimated. We now noted this point in Lines 194-195.

Furthermore, we conducted a benchmark of t_K estimation for the copy number configuration 4:1 and did not observe the issue of overestimation (Figure 1).

Figure 1: The performance of `Butte` on estimating t_K for SCNA state 4:1 with simulated history and SSNV data. We simulated SSNV data for the SCNA state 4:1 for the history shown on the left panel, with varying t_K (0.1, 0.3, 0.5, 0.7) for two pre-defined t_0 (0.1 and 0.3). For a given history, we simulated SSNV data with varying depth of coverage (D), number of available mutations (M) and tumor purity (P). The inferred t_K by `Butte` are shown as standard box plots (each with 100 simulations) where the box refers to the interquartile range, and the true t_K are indicated by blue lines.

RC: *3. What about other possible trajectories to the observed CN profile, involving more complicated structural variants/SCNAs? How will it affect timing estimates? For example, within a region of elevated CN, is CN always constant? If not, how are different possible trajectories to that CN profile accounted for? E.g. What if there's a continuous region with copy number 3-4-3?*

AR: We are grateful for the reviewer's feedback. In our timing analysis, we employed the copy number segmentation results obtained by TitanCNA. We focused on segments with consistent copy number states as predicted by TitanCNA. Timing estimations for gains were conducted independently for each segment, and we did not investigate the interdependence of evolution in distinct genome regions. For instance, in a continuous genomic region with predicted copy number states of 3-4-3, TitanCNA would divide this region into three segments and we would conduct separate timing estimations for each of these segments. Therefore, our method remains 'blind' to the rearrangement structural trajectories of a genomic region, offering timing information solely for the initial and final gains for a segment of interest.

The approach mentioned above might result in very short segments. However, in certain cases, users of our tool, if they strongly believe that two segments with 3 copies, as in the example of 3-4-3, stem from a single gain, have the option to merge the SSNV data for these regions and utilize `Butte`. This strategy proves valuable for timing synchronized gains, such as those observed in chromothripsis events. We have mentioned this aspect in the Discussion section (Lines 438-440).

Finally, even though our method does not explore rearrangement trajectories, in this revised version, we discovered that our timing predictions for complex gains align with the tool ([1]) which employs a graph theory model to reconstruct structural trajectories of specific events (where both SSNV, copy number, and sequence breakpoint data are available). These findings are illustrated in Supplementary Fig. S8.

RC: *4a. Overall clarity of terminology and results. Clarity of the manuscript could be improved, especially with regards to usage of the terms "early" and "late" gains, as well as differentiating between GD and pre- and post-GD gains. Some specifics:*
a. Once methodology is introduced (in the results), can you more explicitly define early and late gains? In the introduction it says late gains are those occurring in last 20% of evolution time and early as occurring in the first 20%. It is unclear how these definitions are applied in practice in the results section. Can the definition be related to the t_0 and t_K estimates, since it presumably depends on them?

AR: In the result section of the revised version, we have defined the threshold for determining early and late gains at 0.2 or 20%, as explained in Lines 218-220. This selection is substantiated by our assessment of precision, true positive, and false positive rates, as detailed in our response to your first comment. We selected this threshold by carefully balancing precision and false positive rates, ensuring a judicious approach to avoid overcalling both early and late events.

RC: *4b. Figure 4 is unclear about when the GD events are occurring. In the corresponding*

parts of the text, the time of GD events in Figure 4 are mentioned. But as far as I can tell, Figure 4 shows all SCNA gains, including non-GD ones (since pre- and post-GD gains are discussed). Could annotations to the figure or clarifications to the text be provided? This is especially important to understanding the results, since there is much discussion about gains that occurred before and after GD. For example, could vertical lines indicate the times for GD and early and late cutoffs, similar to how GD time is shown in Fig S21.

AR: We now added red arrows to indicate GD in Fig. 4a.

RC: **5. Other Figure 4 issues.**

a. It's difficult to see the difference between initiation and arrival in plots like 4A. With the number of events plotted, it is difficult to see which events are connected by the dotted lines that link initiation and arrival. Currently they look like two separate events.

AR: We appreciate the reviewer's feedback. In response, we have made changes by replacing the dashed lines with colored rectangles to connect the initiation and arrival time for an SCNA. This modification enhances the visibility of larger SCNAs or those with a longer evolutionary timeline. We have applied these updates to all figures, including the Supplemental ones, aligning the timing plots with the format seen in the updated Figure 4a.

RC: **5b. Fig 4A left panel for CN is ambiguous with current color choices. I believe right 1/3 of this panel is showing any LOH, but this color is similar to CN = 0. Can LOH color be more obvious difference from those used to indicate CN = 0? Or LOH can be indicated in a different way from CN (e.g. box around affected region).**

AR: We have revised the color scheme for LOH, changing it to gray instead of light blue, and subsequently updated all the affected figures accordingly.

RC: **6. I am not convinced by the conclusion that late gains may contain rate-limiting drivers based on the modeling work. In particular, the mathematical modeling section concludes with the strong statements in lines 301-304, which are not adequately supported by the modeling results. I think there is too much causation implied in these statements, and the logic does not flow. This conclusion is restated throughout (e.g., discussion, abstract, etc.). Even if I'm misunderstanding this conclusion, I think it should at least be rephrased or explained further.**

AR: Thank you for bringing up this concern. We acknowledge that the logical flow in the modeling section of the previous version of the paper did not align well with the conclusion. To address this, we have made the following revisions.

To prevent potential misinterpretation that we asserted all rate-limiting drivers occur late, we have replaced the term **rate-limiting drivers** with **final-expansion drivers**. Our intention

in the modeling section is to demonstrate that alterations appearing late (close to MRCA) might include crucial driver events essential for the tumor's ultimate expansion. This finding highlights the significance of our methodology in identifying late gains, in addition to early gains.

In response to your feedback on the logical flow, we have restructured the mathematical modeling section (refer to pages 14-18). To accurately convey the purpose of the modeling section, its title has been revised to **Mathematical modeling suggests the role of late truncal gains in promoting final expansion**. We have also revised the opening paragraph of this section to succinctly capture the main idea and key insight of the modeling part, which is reproduced below for your reference:

While early genomic alterations garner significant attention for their functional implications, our understanding of the significance of late truncal alterations that emerge in proximity to the MRCA remains limited. Employing mathematical models, we constructed a rationale mentioned in the Introduction: a truncal alteration leading to the final expansion might be situated near the MRCA if the progenitor cell undergoes rapid proliferation upon acquiring this specific alteration. Using GD as an example, as it can occur early or late, we reason that early GD and prior alterations alone are insufficient to drive the final tumor expansion. However, if GD occurs closer to the MRCA, it, along with other late aberrations, can propel the tumor's ultimate growth.

Subsequently, we present two contrasting models: a base model where GD drives the final expansion (indicating that all post-GD gains are neutral), and an alternative model where cells with GD can acquire an additional beneficial post-GD gain. Our analysis reveals that, in the base model, the number of post-GD gains is limited (implying a short time between GD and the MRCA) and follows a distribution centered around zero (indicating GD occurs among the last few somatic copy number alterations close to the MRCA). In contrast, the alternative model predicts a higher abundance of post-GD gains (suggesting a longer period between GD and the MRCA). This finding supports our assertion, reiterated in the second last paragraph of the modeling section, as reproduced below for your reference:

If the identified SCNAs in patient data represent the dominant clone of the entire tumor (a scenario more likely in multi-sampling than single-sampling), our model implies that late clonal gains may harbor drivers for final expansion. The presence of early GD in many patient tumors suggests that post-GD gains might offer added advantages for promoting the final expansion. It's worth noting that our mathematical model doesn't rule out the possibility of late somatic alterations of other types, beyond

copy number gains, driving the final expansion. Thus, investigating late alterations in various forms of somatic changes is an area of interest.

Finally, for a comprehensive perspective, we conducted a brief exploration of the impact of alterations that occur early in truncal evolution, highlighting their role in the tumor's initiation. Specifically, if multiple drivers occur at random, those conferring higher fitness advantages are more likely to occur early. These underscore the value of `Butte` in identifying complex SCNAs with early initiation and late arrival. We have included this discussion below for your reference:

In the context of multiple drivers, previous research (Paterson et al., 2020) indicates that in the random occurrence of multiple drivers, the one enhancing fitness the most is likely to appear early. Our simulations, shown in Supplemental Fig. S30, support this idea. When two drivers occur at equal rates, the one with the greater fitness increase is more likely to emerge first in the initial cell acquiring both drivers. Additionally, the probability of this early appearance increases with the magnitude of the fitness advantage provided by the more beneficial driver. This implies that early gains may involve drivers with significant fitness effects, whereas late truncal gains near the MRCA are particularly relevant for the fitness increase driving the final expansion.

We hope that the revised version adequately addresses your concerns regarding the clarity of the mathematical model interpretations.

RC: *7. Some confidence intervals for timing estimates are very large. Can the authors comment on the magnitude of the uncertainty for the timing estimates? For the tumors analyzed, can they provide summary statistics for the size of the confidence intervals (in units of evolution time spanned).*

AR: We appreciate the reviewer's attention to this issue. Our analysis indicates that a higher number of SSNVs reduces the confidence interval and enhances estimation precision, as illustrated in Supplementary Fig. S10. We suggest employing more than 20 SSNVs for accurate timing estimation of SCNAs, although this might be challenging for small segments or tumors with lower mutation rates. We have addressed these findings in Lines 198-200 and Lines 442-444 in the updated manuscript.

1.3. Minor concern

RC: *1. Running title has hyperlink to unrelated website.*

AR: We express our gratitude to the reviewer for diligently reviewing these points. As a result, we have appropriately addressed the concern by removing the hyperlink.

RC: *2. Define SRE in the introduction at its first mention.*

AR: As there are many abbreviations, we now provided a summary of terminology in Supplemental Table 1. We have also emphasized the abbreviation of sampling relevant expansion (SRE) in lines 76-77.

RC: *3. Line 543. When it is stated in the Methods how to determine GD, it is unclear to me why after clustering inferred times, GD would correspond to the cluster containing 40% of segments. Is this a standard way of identifying GD? If not, how does this compare to other methods for identifying GD?*

AR: It should be noted that multiple clusters of gains can be present during the somatic evolution. We clustered the timing results and used 40% as the cutoff for the dominant punctuated burst of gains (Figure 2A). We regard the averaged timing of all the segments in the corresponding cluster as the timing of GD. We note that this threshold works well for cases with sporadic gains or a secondary burst of gains (Figure 2B). We have expanded the definition of GD and its detection in Lines 243-245 and 619-622, as well as Supplemental Fig. S27. When comparing with Nik-Zainal's work ([2]), we found that our GD timing aligns well with their GD predictions (Supplemental Fig. S8).

RC: *4. What is the "evolution time" mentioned in the intro? This term should be defined.*

AR: Thank you for this comment. We now revised it to be "truncal evolution time measured by truncal/clonal SSNVs".

RC: *5. Why do estimates sometimes have increasing error with increasing read depth (e.g. Fig S3)?*

AR: Thank you for bringing up this insightful observation. Our simulation results indicate that the main approach for enhancing accuracy is to augment the number of SSNVs, as demonstrated in the revised Supplementary Figs. S2-S5. In each simulation iteration, we generate a fresh set of SSNVs according to a multinomial distribution based on the actual allele state distribution (emphasized in Method section Line 594). The discrepancies in allele states resulting from randomly generated mutations contribute to variances in the estimated timing. Notably, when simulations are conducted with a fixed set of mutations (allele states), the variance in timing estimates does decrease as the sequencing depth increases (Figure 3). Please note that the discrepancies observed in the timing estimates from the true values primarily stem from the errors in the allele states.

RC: *6. Is there not difficulty in resolving number of SSNVs at j/N frequencies as copy number N increases?*

AR: We concur with the reviewer that estimating allele state distribution becomes more challeng-

Figure 2: (A) The histogram displays the proportion of segments in timing clusters. (B) Two sample tumors where GD is identified as the timing cluster encompassing over 40% of SCNA segments.

ing with an increase in copy number, given a fixed number of SSNVs. Currently, *Butte* is designed to handle SCNAs up to a total copy number of 7. To ensure accurate performance for copy numbers 6 and 7, we examined the precision of estimating the proportions of SSNVs corresponding to each j/N frequency category (represented by the q vector) in the context of copy number configurations 6:1, 6:2, 7:1, and 7:2, respectively.

Figure 3: To investigate the source of variance in timing estimates, we simulated SSNV data with fixed allele states and manipulated the depth of coverage (D). The plots show the performance of `Butte` in estimating t_K for SCNA state 6:1 with history shown in the left panel. We explored different values of t_K (0.1, 0.3, 0.5, 0.7) with two pre-defined t_0 values (0.1 and 0.3), and the intermediate time stages between t_0 and t_K were evenly distributed. The inferred t_K values obtained by `Butte` are presented in standard box plots, each based on 100 simulations. True t_K values are indicated by blue lines.

Similar to our benchmarking simulations, we generated the true q vector for each copy number configuration based on a known copy number history matrix and specific timing for each stage. Mathematically, this q vector can be computed by the dot product between the history matrix and the time fraction vector. We subsequently simulated SSNV read count data according to the true q , considering a total of 100 SSNVs and an average depth of 150. In each of 200 iterations, we estimate the q vector using the EM algorithm implemented in `Butte`, comparing the estimated q vector to the original one. The performance metrics we employed for this comparison include Cosine similarity and Euclidean distance.

The density plots depicting these two statistics can be observed in Figure 4. Both the Euclidean distance and Cosine similarity density plots indicate that the estimation of the q vector exhibits sufficiently accurate results for copy numbers 6 and 7, given sufficient number of SSNVs and read depth.

RC: 7. *Top left panel on Fig 2 could use more annotation or explanation.*

AR: We have modified Fig. 2 (the top left panel) and its respective figure captions.

RC: 8. *I think the recent Nature Genetics paper by Körber et al (<https://doi.org/10.1038/s41588-023-01332-y>) is relevant to the conclusions and methods discussed.*

AR: We found the mentioned paper highly intriguing, as it aligns with some of the patterns we observed in osteosarcoma. In our revised manuscript, we leverage this paper to explore the potential significance of prolonged evolution after genomic doubling (GD) in Lines

Figure 4: Evaluation of performance in estimating mutation proportions at j/N frequencies for Copy Numbers 6 and 7. The left panel shows cosine similarity, while the right panel displays Euclidean distance, used to compare the estimated q vector with the true one.

429-431.

RC: 9. Line 193: Clarify that “the existing method” is from Cross et al. 2018.

AR: We referenced Cross et al. 2018 in this context to clarify that their method, which focuses on maximizing the joint likelihood of copy number timing, was the one we compared our approach to.

RC: 10. Line 235: “The earlier the timing of GD, the more post-GD CN gains (Fig. 4A).” Fig 4A shows 3 example tumors, so this statement should either be revised to reflect this fact, or reference a figure more appropriate for the generality of this statement.

AR: We should have referred to Fig. 5A in this context. This has been rectified, and we apologize for any confusion caused.

RC: 11. Can equation 2 at line 287 (or equivalently in Methods, Equation 11 (line 613)) be explained and/or derived?

AR: Thank you for your feedback. We have incorporated an explanation elucidating the derivation of Equation 2 (Lines 321-323), which is provided below for your reference:

\bar{S} is thus

$$\bar{S} = \int_0^\infty \mathbb{P}(\sigma_1 > t \mid \Omega_\infty) u_0 dt,$$

where we utilized the formula (see Section V.6 of [3]) to calculate the expected birth time of the first non-extinct type 1 cell. This calculation involved utilizing the tail probability ($\mathbb{P}(\sigma_1 > t \mid \Omega_\infty)$) and multiplying the expected birth time by the passenger mutation rate μ_0 .

RC: *12. Line 273: “Moreover, numerical simulations show that the number of dominant post-GD gains continues to follow a geometrical-like distribution with the mode at zero.” Can the authors support this statement with a reference to a figure?*

AR: We have included a plot (refer to Figure 5 below) in the appendix (Supplemental Fig. S29) to substantiate the mentioned statement.

Figure 5: Histogram of number of dominant post_GD gains. Histogram is based on 10^3 simulations and all simulations used the model parameters: $a_0 = 1$, $b_0 = 0.5$, and $\mu_0 = 0.5$.

RC: *13. Can the authors include citation(s) for the strong and general opening sentence of*

the discussion (line 350)?

AR: We have modified the opening sentence of the Discussion section as follows:

Despite the well-established link between a chaotic genome in tumors with the CIN phenotype and poor clinical outcomes [4], the mechanisms by which specific aberrations contribute to tumor growth remains poorly understood [5].

RC: *14. Lines 399-401: “Specialized pre-selection of SSNVs that faithfully possess a clock-like behavior (Gerstung et al.) is necessary for tumors whose mutagenic processes varies drastically within the studied timeline.” Is that not the case for the tumors studied?*

AR: We appreciate the reviewer’s question, and our considerations are as follows: Tumors undergo multiple mutagenic processes during their development. Our framework operates under the assumption that the relative contribution of different mutational processes remains relatively stable over time within a genomic region, thereby maintaining a consistent mutation rate per base. Specifically, we assume that the mutation rate per base remains constant during the timeline of copy number evolution. If this assumption is violated, the accuracy of SCNA timing derived from mutation data could be compromised.

Research has demonstrated variations in mutagenic signatures between clonal and subclonal lineages, such as the signature of AID/APOBEC cytidine deaminases [6]. However, this concern does not apply to our study because we specifically focused on clonal mutations occurring prior to the establishment of the founder cell lineage. Additionally, a recent analysis by the Quaid Morris group indicated that changes in mutation signatures across the genome are unlikely to be solely attributed to copy number alterations [7]. This finding suggests that copy number changes are unlikely to cause significant shifts in mutagenic processes, thereby validating the assumptions made in our study.

We acknowledge the potential of incorporating mutagenic signatures to select a subset of mutations that accurately reflect the chronological order of events. However, careful method design is essential, as this approach may reduce the pool of available mutations for inference. Therefore, we consider this as a potential enhancement for our timing framework in the future.

RC: *15. Line 48: “The inextricable relation of SCNAs to cancer initiation (Nowak et al. ; Davoli, Xu, et al) and progression (Lukow et al.) has become a consensus in cancer genomics.” I’d be hesitant to use the word “consensus” here, even though it is a well-observed relation. It implies complete agreement in the literature about the primacy of SCNAs in cancer initiation and progression, excluding other important factors like microenvironment, smaller genetic alterations, epigenetics, etc. I don’t think there is*

much doubt that SCNAs play an important role, but I think this phrasing might imply more than is intended.

AR: We appreciate the reviewers' feedback regarding the use of the term "consensus" in relation to the inextricable relation of SCNAs to cancer initiation and progression. We acknowledge that the term "consensus" may imply complete agreement and could potentially overlook other important factors that also contribute to cancer initiation and progression.

Upon careful consideration, we revised the sentence as "The inextricable relation of SCNAs to cancer initiation and progression has become widely recognized in cancer genomics".

RC: *16. The timing ranks in Fig. 6A needs more explanation or visual cues.*

AR: We have incorporated arrows in Fig. 6A to indicate that gene ranking is determined by their timing.

2. Response to Reviewer #2

2.1. General

RC: *In this paper, the authors present an extension of an existing algorithm by Purdom et al. (2013) to predict the timing of copy number gains in cancer tumors, including repeated events at the same locus up to a copy number of 6. This algorithm is applied to several cancer DNA sequencing datasets. For each genomic locus affected by SCNAs, the timings of the first gain and the last gain before population expansion of the sample are estimated. The derived results can provide insight into the dynamics of CN gains in human cancers. However, the progress over existing approaches and findings as well as the validity of the interpretations of the main results remain somewhat uncertain.*

AR: We are pleased that the reviewer acknowledged the insightful nature of our results concerning copy number gains in cancer dynamics. In this revised version, we have emphasized the added value of timing results over copy number alone, assessed the performance in detecting gains with early initiation and/or late arrival, provided detailed interpretations of the mathematical modeling section, and addressed your thoughtful concerns.

2.2. Major Comments

RC: *1. First, the authors should choose what they mean by "arrival time" and maintain this throughout the manuscript. In some places, it is the duration t_K of the interval after the last CN gain before expansion (e.g. caption of Fig. 1), whereas in others it is the time at which the last CN gain takes place, measured relative to birth (e.g. Fig. 1A, Fig. 4), i.e. $1-t_K$.*

AR: Thank you for your suggestion. In the revised version, we have introduced the term **lead time** to represent the period from the last gain to the MRCA (t_K), while we have redefined **arrival time** as the time fraction from time zero to the time of the last gain of a clonal SCNA (i.e., $1-t_K$, as explained in Lines 95-98, updated Fig. 1, and Supplemental Table 1). We have ensured that these terms are consistently used throughout the revised manuscript.

RC: *2. The estimation of t_K is based on the VAF spectrum. In general, the VAF spectrum contains a number of low-frequency variants that occur in the subclonal phase of tumor growth. If present, these could inflate the estimates of t_K . According to the Methods, these variants are filtered out by restricting to clonal variants. However, since the definition of clonal vs subclonal already relies on knowledge of copy number status and sequence of mutational events (to calculate the CCF, l. 466), how could clonality status in the single-sample tumors be reliably inferred?*

AR: To identify clonal SSNVs within a single sample, we utilized a previously published strategy [8]. It is important to note that the detectability of subclonal variants significantly decreases as copy numbers rise, as illustrated in Figure 6, at a constant sequencing depth. For instance, in the scenario where copy number is 5 and tumor purity is 0.6, an SSNV with fewer than 6 mutant reads (out of a total read depth of 50) has a probability of 0.17 of being clonal to the respective sample. This probability increases to 0.3 for copy number 6. Consequently, in cases of high copy number gains and moderate tumor purity, SSNVs identified with a frequency of 0.1 or higher are highly likely to be clonal to the corresponding sample. As a result, this bias naturally limits the inclusion of subclonal variants and ensures a meaningful timing analysis.

Figure 6: The probability of SSNVs being clonal is depicted based on variant allele frequency (VAF) thresholds as total copy number increases. The chart illustrates binomial probabilities, indicating the chance of observing fewer or equal mutant allele reads (up to $VAF \times \text{depth}$) out of a total of 50 reads. These probabilities are calculated considering a lower bound (1 out of total copies) of expected allele frequency if the SSNV is clonal, assuming a tumor purity of 0.6 at the investigated total copy number.

We acknowledge that identifying clonal variants across the entire tumor poses a challenge when only one sample is available. In our analysis, we incorporated cases with multiple tumor samples. Public SSNVs, which are high-frequency variants shared among multiple samples, were utilized as input for the timing analysis. These aspects have been noted in Lines 521-522 of the Methods section.

RC: 3. Another putative confounding factor for the proposed inference is sample purity. Since variants with an alternate read count of less than 3 are usually filtered out of the sequencing data, low purity of a sample means that certain variants will never be seen, even after correcting for purity by dividing the VAF. This would result in a downward bias of t_K . The authors evaluated the effects of purity on the estimation of t_K (Fig. S2,3). In these simulations, was realistic filtering of variants with low alternate read count applied? Do the authors find a correlation between the estimated t_K and purity in the real data? And with the read depth and the number of SNVs?

AR: We appreciate the valuable suggestion provided by the reviewer. In response, we have incorporated the impact of variant filtration due to low depth into our simulation. The updated results are presented in Supplementary Figs. S2-S5. It is important to note that the depth filter influences the outcomes primarily in cases of extremely low depth or purity. In addition, t_K is not correlated with sample purity, read depth or number of SSNVs (see Figure 7).

Figure 7: The averaged t_K is plotted against sample purity, median depth and number of SSNVs, respectively. P values from Spearman correlation tests are shown.

RC: 4. One of the main results is the shift toward small t_K of some putative late driver events such as *MYC* for osteosarcoma in Fig. 4C, placing them close in time to the MRCA of the sample. However, are the small t_K s observed for such events not a necessary consequence of the high CN? In other words, the more events that must take place (i.e. the higher K), the smaller the fraction of time available (t_i) for each of them, since $\sum_i t_i = 1$. This would naturally decrease t_0 and increase $1-t_K$. I do not think that the differences in $1-t_K$ (and t_0) between regions, as shown in Fig. 4C, can be meaningfully interpreted without taking this intrinsic property of the data into account. Similarly, the results shown in Fig. S12 seem expected, since, for example, the 3:1 condition involves one fewer SCNA event than 3:0.

AR: We appreciate the reviewer for this insightful observation. We acknowledge the possibility that higher copy number (CN) states might necessitate a longer duration to establish. This hypothesis, although intriguing, has not been explored in previous research to the best of our knowledge. The lack of appropriate methodologies for measuring the evolution of complex gains has been a significant challenge, which our study aims to improve upon.

To assess whether the estimated timing offers additional insights compared to copy number (CN) data alone, we sought to determine if regions displayed frequent high copy numbers across patients exhibit recurrent early or late gains. To achieve this, we conducted an additional analysis utilizing rank sums, incorporating both timing and copy number data. The raw timing and copy number data were transformed into ranks, and a normalized rank sum was calculated to enable a comparison between these two (detailed in Methods, Lines 627-641).

We observed that the most prevalent SCNAs tended to emerge earlier or establish later during tumor development compared to less common SCNAs, supporting the hypothesis that recurrent SCNAs are likely to be pivotal events driving tumor fitness. However, it is important to note that the correlation between copy number frequency and timing was not consistent across all genomic regions. For instance, in breast cancer, both gains on chromosome 8q and gains on chromosome 1q were equally frequent. Surprisingly, despite their similar frequencies, gains on 1q consistently occurred at earlier stages during tumor development in comparison to 8q gains (Supplementary Figure S7). Recent experiments have demonstrated a substantial fitness advantage associated with 1q gains [5]. Similarly, in osteosarcoma (OS), regions on chromosome 1q and chromosome 7q exhibited gains at a comparable late stage to chromosome 8q, even though they were less frequent in patients. These findings imply that incorporating timing information provides valuable insights beyond what can be gleaned from copy number (CN) data alone.

It is crucial to recognize that although we structured our model to represent copy number gains in discrete steps, real-world scenarios may involve the skipping of certain intermediate steps due to processes such as genome doubling or other mechanisms. Consequently, it is not necessarily accurate to assume that copy number states modeled with more steps take longer to establish than those with fewer steps. Similarly, the timing for copy number states with the same number of steps can vary. For instance, the establishment of the 4:0 state was observed to occur later than that of the 5:1 state (Figure 8). In the revised manuscript, we have emphasized that our comparison between amplified LOH and allele-specific amplifications is confined to cases with the same total copy number.

RC: *5. Regarding the analysis in Fig. 5, I am not sure I follow the conclusion from the mathematical model that late advantageous gains become a rate-limiting event (l. 304). First, it is not clear why Fig. 5A would not be consistent with neutral evolution, since*

Figure 8: Comparisons of the arrival time between SCNA state 4:0 and 5:1. The density curves represent the distribution of arrival time for copy number configuration 4:1 and 5:1 (red) across the analyzed tumors, respectively. 4:0 established significantly later than 5:1 (as indicated by the p value, Wilcoxon rank sum test). Also shown is the Cohen's d as an estimate of the effect size. Vertical red and blue lines indicates the mean values for the two types of SCNAs, respectively. N: number of segments for the corresponding SCNA state.

the number of post-GD gains should be linearly proportional to the mutation rate and the time available for their occurrence, which the figure does show. In truncal evolution, neutral passenger SCNAs, like selected SCNAs, are dominant as soon as they occur. Finally, it appears that neutral evolution is rejected by assuming that u_0 and λ_0 are comparable (l. 272). It is unclear why this is assumed, given how little is known about cancer cell growth and mutation rates. The statement is weaker in the caption of Fig. 5 ("modeling suggests that advantageous CN gains can occur late") and in l. 356, but then it is tautological.

AR: Thank you for bringing up this concern. We agree that the logical flow in the modeling section of the previous version of the paper did not align well with the conclusion. Regarding Fig. 5A, we would like to clarify that in both the neutral and selection models, the number of post-GD gains is directly proportional to the mutation rate and the available time for their occurrence, assuming a constant mutation rate. However, in the neutral model, the time between GD and MRCA is considerably shorter compared to the selection model,

as per our analysis. Consequently, in the neutral model, the number of post-GD gains is limited and follows a distribution with a mode at zero (indicating GD is among the last few somatic copy number alterations close to the MRCA), whereas the selection model predicts a higher abundance of post-GD gains. We have emphasized this point clearly in the revised version of the paper. Specifically, we explicitly state that our mathematical modeling of the mutation mechanism finds partial support from Fig. 5A, which is reproduced below for your reference.

Notably, in both models, post-GD gains are proportional to the mutation rate and time between GD and MRCA, validated partially in Fig. 5A.

We have also revised the opening paragraph of the math modeling section to encapsulate the main idea and key insight of the modeling part, reproduced below for your convenience:

While early genomic alterations garner significant attention for their functional implications, our understanding of the significance of late truncal alterations that emerge in proximity to the MRCA remains limited. Employing mathematical models, we constructed a rationale mentioned in the Introduction: a truncal alteration leading to the final expansion might be situated near the MRCA if the progenitor cell undergoes rapid proliferation upon acquiring this specific alteration. Using GD as an example, as it can occur early or late, we reason that early GD and prior alterations alone are insufficient to drive the final tumor expansion. However, if GD occurs closer to the MRCA, it, along with other late aberrations, can propel the tumor's ultimate growth

About our assumption that u_0 and λ_0 are comparable, we concur that there is limited understanding of the growth and mutation rates in cancer cells. Relying on the limited literature at hand, we cautiously posit that our assumption is reasonable and discussed in Lines 314-315. Specifically, [9] provided an estimation for breast tumors, suggesting a de novo CNA rate of 0.242 per cell division. This implies that provided the death rate of the tumor cells does not significantly approach their birth rate (which is likely to be true when the tumor is under expansion), u_0 and λ_0 can be considered comparable. According to [10], an estimation was made for the ratio of $\frac{\mu_0}{\lambda_0}$ for chronic lymphocytic leukemia, which falls within the range of 0.22 to 1.29. It should be noted that the exome mutation rate, not the CNV rate, was estimated by the authors in [10]. Lastly, it is worth noting that our findings demonstrate that the selection model yields a greater abundance of post-GD gains compared to the neutral model, irrespective of the specific values of μ_0 and λ_0 . However, we acknowledge that if μ_0 is an order of magnitude larger than λ_0 , it becomes considerably more difficult to reject the neutral model in the absence of any information regarding the values of μ_0 and λ_0 .

We hope you find that the revised version adequately addresses your concerns about the clarity of the mathematical model interpretations.

RC: *6. As I understand, each history matrix gives an estimate for each t_i . If a matrix is not invertible, the estimate of t_i is replaced by an upper bound for t_i (only applied to $i=0$ and $i=K$). How are these two qualitatively different quantities (i.e., exact estimates vs upper bounds) combined in diagrams such as Fig. 4B,C?*

AR: In situations where an exact solution is unattainable, we resort to upper bounds as our estimated timing. While these two values are inherently different in nature, employing upper bounds serves the purpose of bridging the knowledge gap regarding the timing of complex gains. By utilizing these mixed timing estimates, we successfully pinpoint punctuated genome doubling (GD) events in the majority of samples. In several instances, the upper bounds of complex gains closely align with the exact timing solutions (of low CN gains) for the GD event. Furthermore, according to our performance evaluations (refer to Supplementary Figs. S2-S5), the median predicted upper bounds for both t_0 and t_K in the high copy number state 6:1 exhibit strong alignment with the actual timing. These justifies the integrated analysis presented in Figure 4B.

Moreover, in the updated Fig. 4C, our objective is to pinpoint recurring early gains (characterized by small t_0) and late gains (indicated by small t_K). Utilizing the upper bounds of these metrics is a conservative approach that can reduce the likelihood of overcalling these events.

RC: *7. To benchmark the method, I would recommend comparing it to existing methods such as that of Greenman et al. (2011) by applying it to the data in that paper and that of Nik-Zainal (2012), which used the same algorithm, to compare the event time predictions.*

AR: We appreciate the reviewer's valuable suggestion. The groundbreaking work of [1] served as inspiration for the development of `Butte`. We agree that the dataset presented in [2] is suitable for evaluating our method. Accordingly, we obtained the corresponding SSNV and SCNA data from the ICGC data portal and applied `Butte` to these datasets. Our analysis revealed strong alignment between the event timing predicted by `Butte` and the findings published in [2]. These results are illustrated in Supplementary Fig. S8 and Lines 203-205 in the revised manuscript.

RC: *8. In addition to applying the method to datasets with existing predictions of CN gain timings, the most compelling benchmark with the available data would be to apply the algorithm to subclones identified in the phylogenetic reconstructions of the multi-region sequencing datasets that carry private SNCAs (Fig. 3). Fig. S8 nicely shows that subclone-private gains in the COAD cohort are indeed predicted to occur later than shared gains on average. Why did the authors not perform the same analysis for all other*

cohorts?

AR: We appreciate the reviewer for bringing up this concern, and we apologize for any confusion caused. In the analysis presented in Supplemental Fig. S11 (original Fig. S8), we conducted assessments for all available tumor types with multi-sampling, not limited to COAD alone. This clarification has been incorporated into the caption of Fig. S9 and highlighted in Line 210 of the main manuscript. We are pleased to confirm that our tool successfully predicted the timing of these subclonal somatic copy number alterations, which were detected later than public SCNAs.

RC: *9. As another way to show the added value of the method, compare the GSEA predictions from Fig. 6A with the ranking of genes simply by their average CN (as opposed to average $1-t_K$). If the enrichment scores are worse, this would be a nice demonstration of the utility of the method. Are p-values or adjusted p-values shown in Fig. 6A? Since several gene sets were tested, a correction for multiple testing needs to be applied.*

AR: We appreciate the thoughtful comment and suggestion. We performed additional GSEA analysis by ranking the genes according to their CN (or segment log2 copy number ratio) instead of the estimated timing in the OS cohort. The enrichment score in the proliferative signaling pathway turned to be smaller than what we observed from the timing (1.28 vs 1.56). The adjusted p values (Benjamini-Hochberg procedure) are 0.3 and 0.15, respectively. We included this result in Supplementary Fig. S19. We now have noted the adjusted p value in the captions of Fig. 6A.

2.3. Minor comments

RC: *10. The derivations of the relationship between SSNV frequencies and CN gains on which the algorithm is based assume that all gains are truncal, resulting in integer N_t values. However, SCNAs that arise (shortly) after expansion would be subclonal in the sample, but could produce similar patterns. In this case, the true history would not match any of the history matrices used. Since the algorithm assumes a fixed set of history matrices and integer N_t values, how will the results be affected if such subclonal gains are present at an SCNA site? For example, in the VAF spectrum shown in Fig. 2, the t_0 state is not at the expected frequency of $5/6=0.83$, but rather at 0.75.*

AR: We appreciate the reviewer's insightful comment. Firstly, we want to highlight that in Fig. 2, the observed allele frequency, which may appear lower than the theoretical fraction, is influenced by tumor purity. We have now emphasized this in the figure caption.

To investigate how the presence of a subclonal SCNA impacts the timing results, we considered a scenario where a subclone acquired an extra copy (with a copy number setting

of 6:1) compared to its parental genome (which is MRCA, 5:1) for a specific genomic region (Figure 9). In this case, the actual copy number profile observed is a mixture between 5:1 and 6:1. For the purpose of this illustration, let's assume that TitanCNA determined the copy number as 6:1 at a single clone solution. For the evolution of the 5:1 state, we set both t_0 and t_K at 0.3 and evenly distributed the time for intermediate stages. For simplicity, we assumed that the additional gain occurred shortly after the MRCA, and the subclone expanded upon the gain, resulting in no new mutations being detected as clonal in the subclone.

For mutations shared between the two clones but with different allele states, we calculated the new expected variant allele frequencies by considering the mixture of the two clones. The expected VAF values deviated slightly from those calculated for the 6:1 state alone. Using mutations simulated from these expected VAF values, we inferred that the SCNA 6:1 established late (see Figure 9), and the initiation time was slightly underestimated.

In our analysis of multi-region samples, we observed a delayed emergence of sample-specific somatic copy number alterations (SCNAs) compared to shared SCNAs (Lines 208-213). It is plausible that these sample-specific SCNAs belong to a subclone within the respective sample, which may be substantial in size, leading to its detection as clonal. Consequently, we have incorporated these findings into the Discussion section, highlighting the possibility that certain late gains could stem from rapidly expanding large subclones shortly after the emergence of the MRCA (Lines 460-463).

RC: *11. In the lower panel of Fig. S2, a non-invertible history matrix is shown, but distributions of exact derived t_K values are given, although in this case only upper bounds can be derived. How can this be?*

AR: We have clarified in Supplemental Figs. S2-S5 that the box plots represent the inferred upper bounds of t_0 and t_K , respectively.

RC: *12. It is not quite clear to me what the authors want to say with Fig. 7. Since the mutation target increases with increasing CN, it is expected that the number of SSNVs correlates with CN. That these SSNVs may be subject to selection is also quite expected.*

AR: Fig. 7 serves two main purposes. Firstly, it confirms the expectation that point mutations accumulate at a rate proportional to their copy number, providing the basis for our methodology. We have included this point in the caption of Fig. 7. Secondly, it communicates a crucial message to the cancer research community, emphasizing the necessity for caution in quantifying evolutionary parameters for subclonal expansion. This caution is essential because SCNA accelerates the accumulation of heterogeneity in point mutations. This point has been discussed in Lines 465-470 of the revised manuscript.

Figure 9: We considered a scenario where the MRCA evolved to a copy number state of 5:1 at fixed time points ($t_0 = 0.3$; $t_K = 0.3$), and an additional gain occurred shortly after the MRCA. We computed mixture allele frequencies and simulated mutation read counts for varying subclonal fractions, allowing us to infer the upper bounds of t_0 and t_K for the 6:1 copy number alteration. The results are presented as boxplots.

RC: *13. The use of the term genome doubling (GD) is confusing. First, there is no definition in the main text of what is the difference between GD and other CN gains. In the Methods, it appears that GD was derived from density-based temporal clustering of SCNAs, and whenever more than 40% were found in one cluster, these SCNAs were considered part of a GD event. The goal of calculating GD appears to be to allow comparison with previous studies that have already assessed the relative rate of SCNA gains and losses before and after whole-GD (WGD). However, in these studies, GD is usually associated with a global, i.e. whole-genome, doubling of ploidy. Why was this definition of GD used rather than a more standard derivation of WGD?*

AR: We do not assume that duplications always cover the entire genome during GD events. Instead, we define GD as a distinct and concentrated burst of gains, encompassing more than 40% of all timed SCNA segments. We have clarified this definition in Lines 618-622 and 243-245. Somatic evolution can involve multiple clusters of gains. To identify the

prominent burst of gains, we clustered the timing results, using 40% as the cutoff (Figure 2A). The averaged timing of segments within the cluster represents the timing of GD. It is noteworthy that this threshold performs well in cases with sporadic or secondary burst of gains (Figure 2B). Furthermore, although we do not assume strict whole genome doubling, the inferred GD timing aligns closely with predictions based on Nik-Zainal's study (Supplemental Fig. S8).

RC: *14. The definition of A_{jk} (l. 153) seems unnecessarily complicated. It is the number of segment copies in stage k that produce the final allele state (i.e. frequency) $a_j = j/N_t$.*

AR: We appreciate the reviewer's valuable feedback and their suggestion to provide a simplified definition of the items within the history matrix. We acknowledge that when presented in conjunction with the context preceding the introduction of the history matrix, the definition can indeed be simplified as proposed. As a result, we have revised our original sentence accordingly.

References

- [1] Greenman CD, Pleasance ED, Newman S, Yang F, Fu B, Nik-Zainal S, et al. Estimation of rearrangement phylogeny for cancer genomes. *Genome Research*. 2012;22(2):346–361. doi:10.1101/gr.118414.110.
- [2] Nik-Zainal S, Van Loo P, Wedge D, Alexandrov L, Greenman C, Lau KW, et al. The life history of 21 breast cancers. *Cell*. 2012;149(5):994–1007. doi:10.1016/j.cell.2012.04.023.
- [3] Feller W. *An Introduction to Probability Theory and Its Applications*. 1971;II.
- [4] Hieronymus H, Murali R, Tin A, Yadav K, Abida W, Moller H, et al. Tumor copy number alteration burden is a pan-cancer prognostic factor associated with recurrence and death. *eLife*. 2018;7. doi:10.7554/ELIFE.37294.
- [5] Girish V, Lakhani AA, Thompson SL, Scaduto CM, Brown LM, Hagenson RA, et al. Oncogene-like addiction to aneuploidy in human cancers. *Science (New York, NY)*. 2023;381. doi:10.1126/science.adg4521.
- [6] Gerstung M, Jolly C, Leshchiner I, Dentro SC, Gonzalez S, Rosebrock D, et al. The evolutionary history of 2,658 cancers. *Nature*. 2020;578(7793):122–128. doi:10.1038/s41586-019-1907-7.
- [7] Timmons C, Morris Q, Harrigan CF. Regional Mutational Signature Activities in Cancer Genomes. *bioRxiv*. 2022; p. 2022.01.23.477261. doi:10.1101/2022.01.23.477261.
- [8] Sun R, Hu Z, Sottoriva A, Graham TA, Harpak A, Ma Z, et al. Between-region genetic divergence reflects the mode and tempo of tumor evolution. *Nature Genetics*. 2017;49(7):1015–1024. doi:10.1038/ng.3891.
- [9] Minussi DC, Nicholson MD, Ye H, Davis A, Wang K, Baker T, et al. Breast tumours maintain a reservoir of subclonal diversity during expansion. *Nature*. 2021;592:302–308. doi:10.1038/s41586-021-03357-x.
- [10] Lee ND, Bozic I. Inferring parameters of cancer evolution in chronic lymphocytic leukemia. *PLOS Computational Biology*. 2022;18(11):896–905. doi:10.1371/journal.pcbi.1010677.

REVIEWERS' COMMENTS

Reviewer #1 (Remarks to the Author):

The authors provided a detailed response to each of my concerns, which were satisfactorily addressed. In particular, they performed significant additional validation to address my concerns about bias in the estimates and classifying early versus late gains. Their revisions to the manuscript improve its clarity and soundness. I have no further major comments.

I have several minor comments below that I request the authors fix before publication, but pending these minor corrections, I find the revised paper suitable for publication.

Minor:

1. Mistake in column label for top table in Fig S18. "Death rate of GD (d0)" should be "Death rate of GD (b0)"
2. Typo "bounnding" in Figs S12, S13
3. Most of the main text uses u for passenger mutation rate but the supplementary table uses μ , as well as line 323 of the main text.
4. What does the vertical height mean in the timing rank plot in figure 6A? Can a label be given?

Reviewer #2 (Remarks to the Author):

I would like to start by commending the authors for their efforts to address the reviewer comments (including from reviewer 1). They have executed the additional analyses with great care and creativity and truly tried to re-evaluate their manuscript based on the feedback they had received. Nice work!

Accordingly, all my comments have been addressed, without raising new concerns. I would just like to add remarks regarding my point 4 about the relationship between CN and t_k . First, I am glad that the authors "acknowledge the possibility that higher copy number (CN) states might necessitate a longer duration to establish", as I would have been very worried if they had claimed that time-dependent processes do not scale with time. The expected distribution of CN states should be roughly following a power law (although it is appreciated that large single jumps in CN can occur at times).

“To assess whether the estimated timing offers additional insights compared to copy number (CN) data alone, we sought to determine if regions displayed frequent high copy numbers across patients exhibit recurrent early or late gains. To achieve this, we conducted an additional analysis utilizing rank sums, incorporating both timing and copy number data. The raw timing and copy number data were transformed into ranks, and a normalized rank sum was calculated to enable a comparison between these two (detailed in Methods, Lines 627-641).”

I think this idea was clever and aimed exactly at describing the intrinsic, i.e. unavoidable, relationship between CN and timing, as higher CN necessitates lower t_0 and t_k (higher $1-t_k$). The description in the Methods section (l. 629-) could be made a bit less ambiguous.

The most important part of this analysis is the new Fig. S7, which shows the relationship between the timing and CN ranks and is highly informative. (The x-axis label is a bit misleading since, if I understood correctly, what is measured is the “CN gain”, not the “frequency of a CN gain”.) Due to the intrinsic correlation between CN and timing, the null expectation for the distribution in the left panel would be a bivariate normal with a positive correlation, while in the right panel we would expect a negative correlation. This is very compatible with what is being observed (more so on the right), which is very good. Without a proper null model of CNVs (which is hard to build, as the authors also noted), we cannot know whether the outlier bins ($Z > 3$ or $Z < -3$) are driven by neutral outlier CN events or by selection. However, the timing analysis now actually gives a clue, since any deviation on the timing axis away from the main bivariate normal distribution on the diagonal is unexpected. In the left panel, this argument clearly applies to the dots belonging to chr1 and chr7, as the authors also pointed out. In the right panel, changes on chr1 and chr8 are compatible with the extreme CNs, which means both measures are equally compatible with either selection or an outlier mutation event, while chr2 is again a more interesting case, since the CN signal is weak, but conditional on CN the timing is unusually early.

Authors' Response to Reviews of

Evolving copy number gains promote tumor expansion and bolster mutational diversification

Zicheng Wang, Yunong Xia, Lauren Mills, Athanasios N. Nikolakopoulos, Nicole Maeser, Scott M. Dehm, Jason M. Sheltzer and Ruping Sun

RC: *Reviewers' Comment*, AR: Authors' Response, Manuscript Text

General Comments

We express our gratitude once more to both reviewers for thoroughly reviewing the revised manuscript. We are pleased that our revisions have successfully addressed the concerns raised by the reviewers.

1. Response to Reviewer #1

1.1. General comment

RC: *The authors provided a detailed response to each of my concerns, which were satisfactorily addressed. In particular, they performed significant additional validation to address my concerns about bias in the estimates and classifying early versus late gains. Their revisions to the manuscript improve its clarity and soundness. I have no further major comments. I have several minor comments below that I request the authors fix before publication, but pending these minor corrections, I find the revised paper suitable for publication.*

AR: We sincerely appreciate your thorough review of our revised manuscript. Your positive acknowledgment of the significant additional validation work and improvements in clarity and soundness is truly encouraging.

1.2. Minor concern

RC: *1. Mistake in column label for top table in Fig S18. "Death rate of GD (d0)" should be "Death rate of GD (b0)"*

AR: Thank you for pointing this out. We have corrected this typo.

RC: *2. Typo “bounnding” in Figs S12, S13*

AR: The identified typo has been rectified. We appreciate you pointing it out.

RC: *3. Most of the main text uses u for passenger mutation rate but the supplementary table uses μ , as well as line 323 of the main text.*

AR: Appreciate your input! The mentioned typo has been rectified, and we have consistently employed u to represent the mutation rate throughout the paper.

RC: *4. What does the vertical height mean in the timing rank plot in figure 6A? Can a label be given?*

AR: In the legend of Figure 6A, we have added a note specifying that "The height of each bar corresponds to the arrival time."

2. Response to Reviewer #2

2.1. General

RC: *I would like to start by commending the authors for their efforts to address the reviewer comments (including from reviewer 1). They have executed the additional analyses with great care and creativity and truly tried to re-evaluate their manuscript based on the feedback they had received. Nice work!*

Accordingly, all my comments have been addressed, without raising new concerns. I would just like to add remarks regarding my point 4 about the relationship between CN and t_k . First, I am glad that the authors “acknowledge the possibility that higher copy number (CN) states might necessitate a longer duration to establish”, as I would have been very worried if they had claimed that time-dependent processes do not scale with time. The expected distribution of CN states should be roughly following a power law (although it is appreciated that large single jumps in CN can occur at times).

“To assess whether the estimated timing offers additional insights compared to copy number (CN) data alone, we sought to determine if regions displayed frequent high copy numbers across patients exhibit recurrent early or late gains. To achieve this, we conducted an additional analysis utilizing rank sums, incorporating both timing and copy number data. The raw timing and copy number data were transformed into ranks, and a normalized rank sum was calculated to enable a comparison between these two

(detailed in Methods, Lines 627-641)."

I think this idea was clever and aimed exactly at describing the intrinsic, i.e. unavoidable, relationship between CN and timing, as higher CN necessitates lower t_0 and t_k (higher $1 - t_k$). The description in the Methods section (l. 629-) could be made a bit less ambiguous.

The most important part of this analysis is the new Fig. S7, which shows the relationship between the timing and CN ranks and is highly informative. (The x-axis label is a bit misleading since, if I understood correctly, what is measured is the "CN gain", not the "frequency of a CN gain".) Due to the intrinsic correlation between CN and timing, the null expectation for the distribution in the left panel would be a bivariate normal with a positive correlation, while in the right panel we would expect a negative correlation. This is very compatible with what is being observed (more so on the right), which is very good. Without a proper null model of CNVs (which is hard to build, as the authors also noted), we cannot know whether the outlier bins ($Z > 3$ or $Z < -3$) are driven by neutral outlier CN events or by selection. However, the timing analysis now actually gives a clue, since any deviation on the timing axis away from the main bivariate normal distribution on the diagonal is unexpected. In the left panel, this argument clearly applies to the dots belonging to chr1 and chr7, as the authors also pointed out. In the right panel, changes on chr1 and chr8 are compatible with the extreme CNs, which means both measures are equally compatible with either selection or an outlier mutation event, while chr2 is again a more interesting case, since the CN signal is weak, but conditional on CN the timing is unusually early.

AR: We appreciate the feedback concerning Supplementary Figure S7 (now indexed as Supplementary Figure S14), specifically the representation of the x-axis, which denotes the CN gain for each genomic bin. In response to this observation, we have revised the axis label to accurately reflect this clarification.

We express our gratitude to the reviewer for acknowledging the potential of the timing axis in facilitating the identification of oncogenic SCNAs. We concur with the reviewer's observation that genomic regions, exemplified by Chr2, exhibiting the pattern of 'once it gains, it gains early' are promising candidates. A more in-depth analysis of the deviation from the null distribution on the timing axis could prove valuable in exploring these aspects further. These insights suggest potential directions for future follow-up investigations stemming from the current study.